# Inductive Transformers: How Language Models Form Concepts, And How to Make Them Even Better At It

**Ben Vigoda**[*]**, Thomas Rochais**

## Abstract

We derive transformers from more a more foundational underlying inductive bias. This new understanding enables us to design transformers with tighter conceptual organization, greater conceptual control, and higher levels of conceptual abstraction. We explain the approach and give an illustrative example simulation.

We show that training data can be replaced or augmented by making modest design modifications to the transformer's activation functions and connectivity. We show how to generate synthetic training data that can be used to train inductive bias into a transformer before or in concert with natural language training data.

## 1 Introduction and Prior Art

Our goal is to create language models that learn better organized concepts, more controllable concepts (Wang et al., 2023; Meng et al., 2023; Hernandez et al., 2023), and more abstract concepts. This could in turn help unlock a range of enhanced abilities including better causal reasoning, iterative experimentation, longer range planning, longer chains of reasoning, curiosity, and introspection.

Causal reasoning requires the ability to intervene between connected concepts in a model (Pearl, 1995). Iterative experimental design and interpreting results requires the ability to structure latent concepts to create hypotheses and explain observed data (Lu & Zhang, 2022). Long-range plans and chains of reasoning require the ability to compose sequences of latent concepts (Lake et al., 2017; Oh et al., 2017; Shinn et al., 2023). Curiosity consists of noticing which data is explained well by existing concepts and which data requires further conceptual structures to explain away (Mazzaglia et al., 2022; Chen et al., 2022; Pearl, 1988; Peterson et al., 2019). More speculatively, introspection of ones own reasoning may benefit from concepts that are well-organized and uncertainty that is well characterized.

How can we achieve AI models with deeper conceptual abstractions and greater conceptual clarity? Frontier models may continue to push the envelope with greater quantities of training data and parameters, while also requiring commensurate increases in training compute costs (Amodei & Hernandez, 2018). Human learners, however, are able to learn deep abstractions and great conceptual clarity with at least four orders of magnitude less training data compared to current state-of-the-art models (Frank, 2023).

Reinforcement learning with small but high-quality data sets and improved loss functions continue to be an important path forward (Knight, 2023; Thomaz et al., 2006). This is analogous to tutoring children, but children without significant tutoring are still able to learn very effectively (Gopnik et al., 1999).

Much current effort involves expanding to additional data modalities (e.g. video) (Sun et al., 2019). Extraordinary humans like Helen Keller, however, achieve the highest levels of abstraction and conceptual organization without any visual or auditory inputs (Herrmann, 1999).

Inductive bias is a key under-exploited approach for improving models (Goyal & Bengio, 2022), and many have pointed out the importance of introducing inductive bias into models (Mittal et al., 2022; Goyal & Bengio, 2020; Lamb et al., 2021; Gruber, 2013). Well-designed inductive bias enhances the

---

[*]Direct correspondence to ben@benvigoda.com.

predictive power of a model by shaping the model to be a more likely fit for high-quality data, and a poorer fit for low-quality data (MacKay, 2003). Examples in language models would be a preference for computer code that is syntactically correct or for mathematical proofs that are logically valid. Examples in human learning are exemplified by the individuals such as John von Neumann who exhibited a strong predisposition for learning and manipulating mathematical concepts.[1]

Inductive bias adds constraints that a model could eventually learn with enough time, compute, and high-quality data (Welling, 2019). The additional constraints, however, reduce the degrees of freedom that need to be explored while learning, and by extension during inference.

In fact, the right inductive bias can be used as a substitute for orders of magnitude more high-quality training data. For example, "a controller optimization system equipped with differentiable simulators converges one to four orders of magnitude faster than those using model-free reinforcement learning algorithms" (Belbute-Peres et al., 2018), and on Emotion and AG News benchmark tasks, models pretrained on entailment data outperformed models five hundred times larger (Ge et al., 2023).

(Sartran et al., 2022) modify the existing TransformerXL (Dai et al., 2019) to create "grammar transformers" which tag parts of speech within sentences and then dynamically mask the attention matrices based on these tags. They do not focus beyond the limits of each sentence, and only address their inductive bias to the attention mechanism, not to the entire model. That said, on several benchmarks they demonstrate equivalent performance to models five hundred times larger than their own. This provides compelling evidence for the effectiveness of inductive bias at scale.

The opposite of inductive bias is to remove constraints from the model and simply use more training data. For example, (Liu et al., 2021) replaced attention layers with more generic fully connected perceptron layers, but recovered equivalent performance by increasing the size of the training set.

Transformer models are often summarized as a conditional distribution of the next token given previous tokens, $p(t_{i+1}|t_i, \ldots t_{i-N})$ where $N$ is the context window length. This sometimes gets reinterpreted in the popular imagination as implying that the transformer is simply learning to parrot back sequences of words that it has seen before, i.e. it is "fancy auto-complete" (Marcus et al., 2023). As we will see, there is more structure in these models than implied by this articulation (Veres, 2022).

That said, today's "vanilla" transformers seem to organize internal concepts somewhat loosely and unreliably unless extreme quantities of data and reinforcement are applied (compared to human learning). A great deal of research has been dedicated to understanding how information is encoded within deep learning networks. For example, convolutional networks trained on images have been shown to encode increasing abstraction in increasing layers of the network. This can be demonstrated by stimulating neurons at different layers and observing the images that the trained network outputs (Bau et al., 2020). Looking for similar patterns in transformers has been less conclusive (Clark et al., 2019). "BERTology has clearly come a long way, but it is fair to say we still have more questions than answers about how BERT works" (Rogers et al., 2020). Current approaches have been primarily limited to token semantics, sentence syntax, co-reference and parts of speech Clark et al. (2019) as well as post-facto investigation of small circuits that emerge from training toy models (Elhage et al., 2021).

Designing inductive bias for better and broader conceptual organization requires a modeling prior (Frankle & Carbin, 2019). Goyal and Bengio propose principles for additional inductive bias (Goyal & Bengio, 2022). Paraphrasing their list, (1) knowledge is factorized in terms of abstract variables and functions, (2) high-level variables play a causal role and learn representations of latent entities/attributes, (3) changes in distribution are due to causal intervention and are localized, (4) short causal chains of concepts at higher concept levels organize groups of lower level concepts in order to span very complex explanations or plans, and (5) top-down contextual information is dynamically combined with bottom-up sensory signals at every level of the hierarchy of computations

---

[1]Peter Lax wrote, "... had he lived a normal span of years, he would certainly have been a recipient of a Nobel Prize in economics. And if there were Nobel Prizes in computer science and mathematics, he would have been honored by these too..." (Neumann & Redei, 2005). By age six, he could divide two eight-digit numbers in his head (Schneider, 2015; Henderson, 2007). The Nobel Laureate Hans Bethe said, "I have sometimes wondered whether a brain like von Neumann's does not indicate a species superior to that of man" (Macrae, 1992; Blair Jr, 1957).

relating low-level and high-level representations. Our family of inductive transformers aspires to strongly adhere to these desiderata.

We start with the question, "What is the generative statistical model such that recursive marginalization of the model is in tight equivalence with the calculations performed by inference in a vanilla transformer?" We show that understanding transformers from this perspective can provide a foundation for the design of new inductive bias, yielding *inductive transformers*.

## 2 THE INDUCTIVE TRANSFORMER MODEL

To focus on designing inductive bias into the model, we want to write down the model structure first without worrying about inference or data. Once we define the model, we will define inference employing marginalization, as well as implement learning with back-propagation. By focusing first on the model in isolation, the inductive bias in the model is more evident.

We expect a large language model to estimate uncertainty about underlying discrete variables. Why? Language understanding and generation systems must solve an inverse problem. I transform my concepts into speech when I communicate to you. If you would like to guess what concepts I was thinking, so that you can consider and reply appropriately, you must (approximately) invert my speech back into concepts. This is the foundation of digital and symbolic communications systems going back to Shannon (Shannon, 1948). The mapping from concepts to speech is many-to-many, so you have an inherently under-determined problem to solve, which by its nature requires representing the uncertainty of competing interpretations of the data.

Perhaps the simplest building block that you could employ to model my thought process would be: (1) I generate a single token from a categorical distribution $\pi_T$ over tokens, *and* (2) I choose a $\pi_T$ from which I will generate my next token, by sampling from a distribution $\pi_Z$ over $\pi_T$'s. Then I repeat this simple "production" over and over again. In other words, you model my mind as being made of an incredibly rudimentary grammatical production, but with an enormous number of such productions, trained and wired together in intricate ways. We are not saying that language models are simply *sampling* from a generative grammar. On the contrary, during inference activations represent uncertainty with continuous values. As well, productions are tiled together at enormous scale, with each trained to have its own weights. Our detailed choices in the basic building block (ie. "production") are how we design the inductive bias. Let's investigate in more detail.

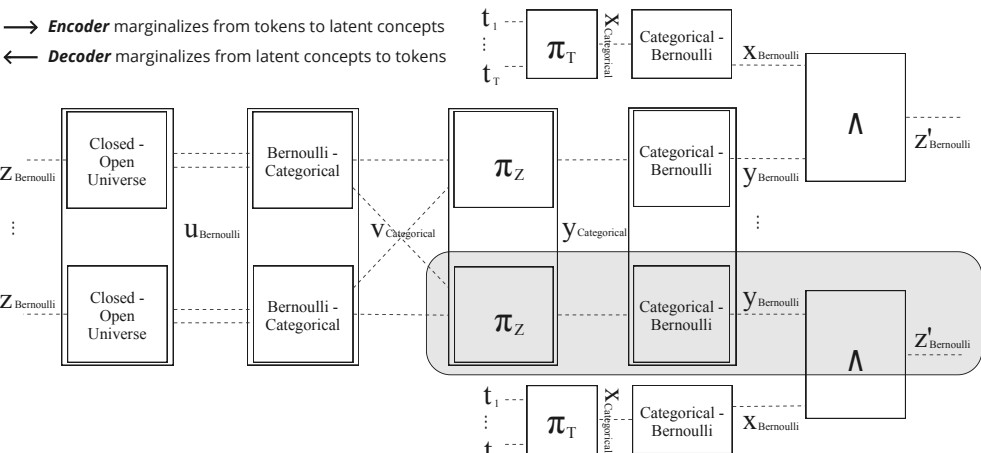

Figure 1: A single layer of the inductive transformer production represented as a factor graph.

To understand the underlying production for a vanilla transformer, we step through a sequence of sampling operations representing a single path through one decoder layer. The $\wedge$ on the right side of figure 1, is an "AND" activation function, detailed in appendices 46,B.6, and D. When activated

by $z'$, it must activate both of its child variables $x$ *AND* $y$. $x$ then activates $\pi_T$ which is a categorical choice over tokens $t \in T$. When activated by the $\wedge$, $\pi_T$ "rolls a die" to choose a token. Because it generates a "terminal symbol", $\pi_T$ corresponds to the residual (or "skip") connections in the vanilla transformer which connect internal layers to a position in the data (more on this in appendix A.1). The $\wedge$ also activates the child variable $y$ which activates $\pi_Z$. When activated by the $\wedge$, $\pi_Z$ chooses an $\wedge$ in the layer below. We will discuss the closed-open universe and categorical-Bernoulli factors later, and in full detail in appendices B.1 and B.4.

In summary, this simplified production generates one token *and* chooses to activate one of the productions in the layer below. A path through multiple layers of strongly connected productions can generate a coherent distribution over possible token sequences. We will refer to this kind of strongly connected sub-network as a "*concept*".

There are many directions for variation and expansion of inductive bias in the transformer: (1) The definitions of $\pi_T$ and $\pi_Z$ can be expanded as shown in appendix C.1 when we incorporate (relative) position to implement an attention mechanism which closely resembles the XL-Transformer Dai et al. (2019). (2) Because it makes a categorical choice over tokens, this production generates embedding vectors that represent one token per dimension, but this will be expanded to represent vanilla sparse semantic distributions in appendix E. (3) The production could allow for each $\pi_Z$ to (jointly) choose *two* or more productions and/or each $\pi_T$ to choose two or more tokens. (4) An inductive bias for context-free grammar could be designed in order to prefer syntactically correct computer code. Perhaps a production could also be designed to favor the generation of formally correct statements and/or steps in an axiomatic system such as Zermelo–Fraenkel set theory (Hrbacek & Jech, 2017). (5) Other biases could be introduced by making use, for instance, of the ontological architectures explored in Gruber (1993; 1995). For space and clarity, we initially content ourselves with presenting our methodology for designing inductive bias with the simplified example in figure 1. Remarkably, once we derive inference in a model made of a large number of these productions tiled together, the vanilla transformer essentially pops out of the derivation. This provides clear opportunities to both tighten and expand inductive bias in transformers by modifying the production and repeating the derivation.

Although our simple production resembles probabilistic generative grammars which have generally been used to model the generation of a sentence, given the penchant in biological evolution for the preservation and reuse of existing evolved structures, we see no reason to presume that this production would stop being used at the punctuation mark. The production seems to naturally fit what humans call outline form for books, composition forms in music such as the sonata (Lerdahl & Jackendoff, 1983), and the hierarchical categories and attributes expressed in symbolic systems such as the Dewey decimal system and relational databases where a particular $\pi_T$ can be viewed as modeling a relation between a subject $\pi_T$ above and an object $\pi_T$ below.

In table 1, we compare the vanilla transformer (Vaswani et al., 2017) to the inductive transformer layer by layer.

| Layer Type | Vanilla Transformer | Inductive Transformer |
|---|---|---|
| Self-attention | $y_i = \sum_j \omega_{i,j} v_j$, where $\omega_{i,j} = \text{Softmax}(q_i k_j^T)$ | We do not modify the attention layer, we derive it as marginalizing a statistical production. See appendix C.1 |
| Add & norm | Sum the residual connections and the outputs from the attention layer below. | Marginalization of the "∧" sums the the token activations output from $\pi_T$ with the attention activations from $\pi_Z$. See appendix D |
| Residual connections | Connections between the input data and internal layers | Generative production where every non-terminal must generate at least one terminal. See appendix A.1 |
| Encoder-decoder connections | Output of the final encoder layer is provided to every decoder layer | When we detail forward and backward marginalization in the model, we will see that each layer of the encoder should provide marginals to the corresponding decoder layer. See appendix A.2 |
| Feed-forward | Columnar MLPs possibly learning to approximate the corresponding activation functions in the inductive transformer | Marginalize the posterior log probability of the categorical-Bernoulli, open-closed universe, and Bernoulli-categorical factors. See section B.1 |

Table 1: Comparison of Vanilla and Inductive Transformer Layers

As we discuss in table 1 above and detail in appendix C.1, we strive for a close correspondence between the equations for the vanilla attention mechanism and the equations we derive by marginalizing our attention production.

Similarly, the correspondence between the ∧ factor and the add & norm layers in the vanilla transformer is strongly suggested by the fact that these layers are where the residual connections get combined with the activations from the layer below. Furthermore there is a close mathematical correspondence between the implementation of the ∧ in the log probability domain and the add & normalization operation (see further details in appendix C.1).

Much is therefore the same. Where do the inductive and vanilla transformers differ? There is one difference in how the encoder of the inductive transformer should connect to the decoder, where vanilla transformers likely must learn to convey this same information through the residual stack. See appendix A for more details.

More substantially, let us look at the feed-forward layer. In the vanilla transformer, the feed-forward layer applies the same operation (essentially a multi-layer perceptron) to each embedding vector coming into it. The vector at each position is processed through its own perceptron independent of the vectors at other positions, but the same weights are employed at every position – these independent operations are identical to one another. Similarly, when we derive the inductive transformer, we find by process of elimination that the closed-open-universe factor and the Bernoulli-to-categorical factor (with its subsequent layer norm) must be somehow performed by the feed-forward layer in the vanilla transformer in order for there to be a tight correspondence between the two approaches. Miraculously, when we implement inference as marginalization on a model comprised of layers of productions, the same independence of columns as well as the subsequent layer add & norm falls out of the inductive transformer derivation. In essence we recover the exact same conditional independencies in the factor graph for the inductive transformer as are present in the vanilla transformer, and they fall out not as the result of tinkering, but as the result of theory where our guiding force was simply to marginalize the production while also optimizing the $O()$ to avoid exponentially complex computations!

This is highly suggestive of a strong correspondence. There is an important difference between the approaches, however. In the inductive transformer we are precisely defining the functions for

our "feed-forward" layer to implement B. In the vanilla transformer these same functions must be learned from data. This suggests that perhaps we ought to pretrain the feed-forward layers of vanilla transformers with synthetic data designed to teach them how to be an open-closed-universe-Bernoulli-to-categorical factor. Conversely, as we relax this layer of the inductive transformer back to being a layer of tied multi-layer perceptrons (MLPs), we recover the vanilla transformer.

## 3 INFERENCE IN THE INDUCTIVE TRANSFORMER

In this section, we will start to see that we can understand inference in a transformer not just as predicting the next token given previous tokens, but as inferring "forward" into a latent space of concepts and then "backwards" through concepts to predict tokens in the token window. The inductive transformer is a more focused version of the vanilla transformer, and will therefore generalize similarly. The time and space complexity is identical.

Determining the latent concepts given the input data is, in general, an under-determined inverse problem. When the probability distribution of a model can be represented by a directed acyclic graph, however, forward-backward marginalization of the model to compute concept likelihoods is exact and computationally efficient Yedidia et al. (2003).

Although our highly connected multilayer neural network may appear to be a cyclic graph, in fact the model represented by concatenation of our productions is a tree. It is only the transformation from an open-universe model to a closed-universe model, discussed in detail in appendix B.1 that makes the model appear to have loops.

The conditional distribution for the inductive transformer decoder in figure 1 is,

$$p(z|u)p(u|v_{\text{Categorical}})p(v_{\text{Categorical}}|y_{\text{Categorical}})p(y_{\text{Categorical}}|y)p(t|x_{\text{Categorical}})$$
$$p(x_{\text{Categorical}}|x)p(x,y|z')p(z'). \tag{1}$$

where $\pi_T = p(t|x_{\text{Categorical}})$ and $\pi_Z = p(v_{\text{Categorical}}|y_{\text{Categorical}})$.

We call $p(x_{\text{Categorical}}|x_{\text{Ber}})$ and $p(y_{\text{Categorical}}|y_{\text{Ber}})$ "Bernoulli-to-Categorical" factors. We represent Bernoulli variables with the subscript "Ber" or with no subscript. We use the subscript "Categorical" to denote Categorical distributions which collect multiple Bernoulli variables into a single joint variable across a layer of activations. This turns out to be important in order to avoid exponential computational complexity in certain layers. See appendices B.2 and B.4 for more details.

As we input a prompt, rightward marginalization in figure 1 computes activations at each layer of the *encoder*. Conditioned on the concepts activated in the encoder, leftward marginalization through the factor graph infers the *decoder* activations. During leftward marginalization, tokens are sampled from the probabilities (activations) in the $\pi_T$'s.

Now we derive the equations for marginalizing the inductive transformer. A transformer architecture may contain an encoder and/or a decoder. We start with the decoder. Inference in a layer of the decoder marginalizes the conditional distribution in equation 1. To massively reduce the computational complexity of the marginalization, we push each summation as far to the right as we can,

$$p(z) = \sum_u p(z|u) \sum_{v_{\text{categorical}}} p(u|v_{\text{Categorical}}) \sum_{y_{\text{categorical}}} p(v_{\text{Categorical}}|y_{\text{Categorical}})$$
$$\cdot \sum_y p(y_{\text{Categorical}}|y) \sum_{x_{\text{categorical}}} p(t|x_{\text{Categorical}}) \sum_x p(x_{\text{Categorical}}|x) \sum_{z'} p(x,y|z')p(z'). \tag{2}$$

Some of the conditional distributions in this equation are,

$$p(x,y|z') = \delta(z'_{\text{Ber}} - \wedge(x_{\text{Ber}}, y_{\text{Ber}})), \tag{3}$$

$$p(v_{\text{Categorical}}|y_{\text{Categorical}}) = W_{v,y}, \tag{4}$$

$$p(t_{\text{Categorical}}|x_{\text{Categorical}}) = W_{t,x}. \tag{5}$$

where $W$'s are learned weight matrices. The *encoder* marginalizes in the opposite direction of the decoder, with conditional distributions that impose the same joint constraints on adjacent variables. Detailed and pedagogical equations for each summation are provided in appendix B.

## 4 ILLUSTRATIVE EXAMPLE

Before concluding, let's zoom into a tiny component of a larger inductive transformer to see the real-world operation in detail. Our focus is on demonstrating the operation of the underlying circuits in the inductive transformer.

### 4.1 MODEL WEIGHTS AND ACTIVATIONS

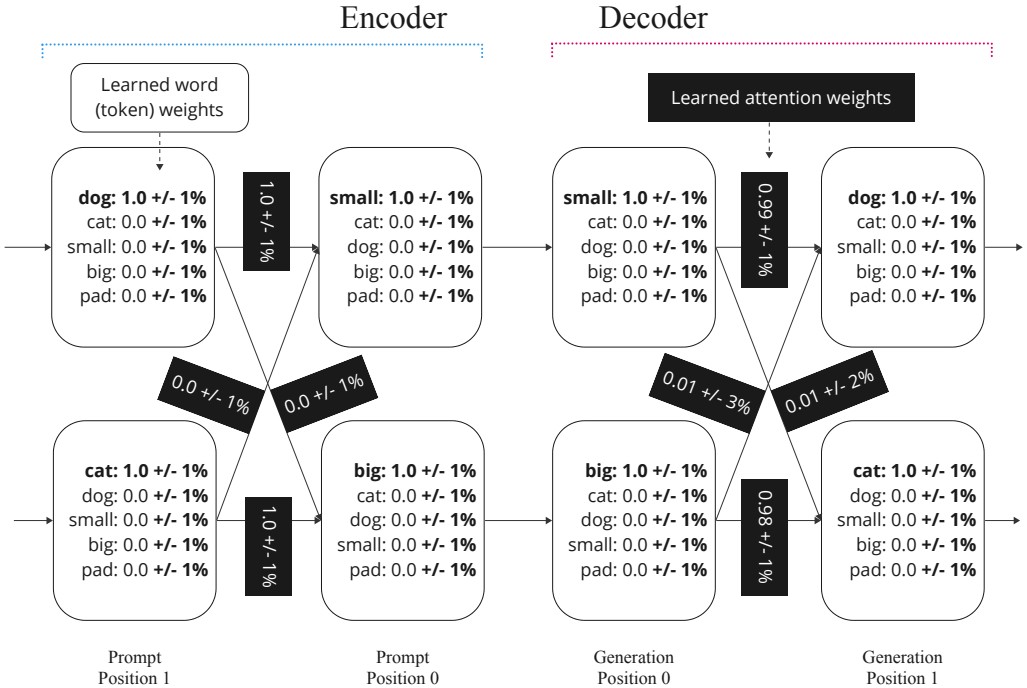

Figure 2: Learned Weights in the Inductive Transformer. The learning is highly reproducible. In a hundred different learning runs, the variance of each learned weight is generally less than 1%. The attention $\pi_Z$ weights are in white with black background while the token $\pi_T$ weights are black on white, next to their corresponding vocabulary words.

We successfully train the inductive transformer even as the data set size scales to zero. This lets us zoom in on a two-layer section of the model with layer width of two. We use a maximally sparse embedding representation described in more detail in appendix E. This highly minimized instance of the inductive transformer generates a single token per production and therefore a single token per layer. In other words, $P$ tokens in the data window needs to be explained away by $P$ layers in this version of the inductive transformer. If we desire a model architecture that can compress more tokens into fewer layers, we adjust the production so that a single layer is able to generate more than a single token.

The model was implemented in PyTorch and trained with back-propagation using the Adam optimizer (Paszke et al., 2019; Kingma & Ba, 2017) on a single NVidia V100 GPU (although a small CPU would have been entirely adequate). The training data were the sentences 'big cat.' and 'small dog.'. In figure 2 we see each learned weight with its variance across a hundred training runs.

## 4.2 PROMPTING AND GENERATION

How does prompting and generation happen in the inductive transformer? When we prompt the model with the word "big", the system generates the phrase "big cat". When we prompt the model with the word "small" it generates the phrase "small dog". Given its hierarchical categorical nature, there is a sense in which the encoder conceptualizes "small" and "big" as kinds or modifiers of "dogs" and "cats".

| Prompt | Generation | Percentage of Generations |
|--------|------------|---------------------------|
| "big"  | "big cat"  | 100% |
| "big"  | "big dog"  | 0% |

| Prompt  | Generation  | Percentage of Generations |
|---------|-------------|---------------------------|
| "small" | "small dog" | 100% |
| "small" | "small cat" | 0% |

## 4.3 IDENTIFIABILITY

The inductive transformer strongly organizes the concepts it learns; It organizes its concepts (1) the same way as the model that generated its training data, and (2) the same way every time. This novel training repeatability is a consequence of the strong inductive bias.

In our context, "identifiability" means the ability for a learning model to receive instructional data from a teacher model, and repeatably learn to mirror the teacher model's structure. To determine if our model is identifiable in this sense, we follow these steps:

1. Create a *forward model* with weights set to particular values.
2. Generate tokens (*generated data*) from this forward model.
3. Copy the forward model to create an *inverse model* with randomly initialized weights.
4. Use the generated data from the forward model to train the inverse model.
5. Compare the (learned) weights in the inverse model to the weights in the forward model. If the weights in the inverse model converge to the same values as the corresponding weights in the forward model, then we say that the model is identifiable.

We see in figure 2 that when repeatedly trained on the same data, the inductive transformer repeatably learns to position the same concepts in the same places within the model. This is repeatable with only a small nudge in one corner of the model to break symmetry. On larger data sets, longer range correlations in the data ensure this symmetry breaking. This suggests the possibility of designing large language models that repeatably learn what we want to teach them.

The identifiability in the inductive transformer is also reminiscent of the fact that for a wide range of concepts, different humans from diverse backgrounds learn to localize particular concepts at the same positions in their brains (Huth et al., 2016; Li et al., 2023; Geva et al., 2021; Merlin & Toneva, 2022).

## 4.4 CONTROLLABILITY

Now we demonstrate that we can delete concepts in the inductive transformer, so that the model will no longer generate text from those concepts. Suppose the section of the model shown in figure 2 was trained with the three sentences "big cat", "big dog", and "small dog", so that while everything else stays the same, the $\pi_Z$ in the 'big' production learns weights $[0.5, 0.5]$ and when prompted with the word "big", the model generates outputs:

| Prompt  | Generation  | Percentage of Generations |
|---------|-------------|---------------------------|
| "small" | "small dog" | 100% |
| "small" | "small cat" | 0% |

| Prompt | Generation | Percentage of Generations |
|--------|------------|---------------------------|
| "big"  | "big cat"  | 50% |
| "big"  | "big dog"  | 50% |

Table 2: After training, the model accurately reflects the training data.

If we lesion the connection between the "big" production and the "dog" production, then the model can only say "big cat" and "small dog", and will no longer say "big dog":

| Prompt | Generation | Percentage of Generations | | Prompt | Generation | Percentage of Generations |
|---|---|---|---|---|---|---|
| "small" | "small dog" | 100% | | "big" | "big cat" | 100% |
| "small" | "small cat" | 0% | | "big" | "big dog" | 0% |

Table 3: Prompted generations from the model where we broke the connection between "big" and "dog".

This demonstrates that the inductive transformer can learn causal relationships between connected sub-networks of productions. We define a "concept" as a sub-network that can generate many different but synonymous token sequences (e.g. "tiny canine"). Given the very close mathematical similarity between inductive and vanilla transformers, it seems very likely that vanilla transformers also form these kinds of concept sub-networks. Although concepts may not be highly localized or organized in the vanilla transformer, they could increasingly be so if we add further inductive bias. Furthermore this suggests that the "emergent" capabilities of large language models as they scale (Wei et al., 2022) may be the result of adding additional "layers" of concepts that provide higher levels of abstraction.

Model controllability has practical implications. It could, for example, make it safer and simpler to share learned weights between models. With concept controllability, after training models on new data and/or reinforcements, people or organizations who exchange weight updates could verify and control the concepts that are being shared. Controllability could also make it possible to edit concepts directly in a model rather than spending far greater effort to review and edit training data. In fact, by linking particular conceptual pathways in the model with particular sections of text, inductive transformers could also be used to help scrub data and to identify intellectual property. Concept controllability could also be utilized to help enhance AI alignment.

## 5 DISCUSSION

This paper offers the following contributions: (1) We provide the first demonstration of causal intervention in a transformer model. For example, we show how to delete specific concepts by deleting specific sub-networks. (2) We design a transformer that successfully learns even as the data set size scales to zero. (3) We design a transformer such that the concepts it learns are localized within identifiable sub-networks. (4) We show that the feed-forward layers of a vanilla transformer learn underlying functions that can instead be derived analytically. (5) We derive from first principles why the multi-layer perceptrons in the feed-forward layer of the vanilla transformer are factored the way they are. (6) We show that the connectivity from the encoder to the decoder in the vanilla transformer is not correct and how to fix it.[2] (8) We derive the testable prediction that training data with a particular inductive bias can help unlock a range of important new abilities for large language models, including curiosity. One could generate synthetic data from, for example, the model we described here, and use this synthetic data within the overall mix of training data for a foundation model (Akyürek et al., 2020). (9) We show that this inductive bias training data can be replaced or augmented by directly designing the inductive bias into the activation functions and connectivity of the model. (9) We mathematically define concepts, and explain why scaling up models yields greater conceptual abstraction. We suggest that deeper abstractions manifest as "emergent" capabilities.

---

[2] Personal communication with researchers at DeepMind confirms that they have recently evolved towards the connectivity predicted by our inductive transformer

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

## A    CONNECTIVITY IN THE INDUCTIVE TRANSFORMER

### A.1    RESIDUAL CONNECTIONS IN VANILLA TRANSFORMER → REDUCED PRODUCTIONS IN THE INDUCTIVE TRANSFORMER

Transformer architectures (and other deep learning architectures) utilize residual (ie. skip) connections. Residual connections have been empirically demonstrated to smooth the fitness landscape helping back-propagation converge:

> "We find that network architecture has a dramatic effect on the loss landscape. Shallow networks have smooth landscapes populated by wide, convex regions. However, as networks become deeper, landscapes spontaneously become 'chaotic' and highly non-convex, leading to poor training behavior... Skip connections cause a dramatic 'convexification' of the loss landscape" (Li et al., 2017).

When learning the productions for an arbitrary generative grammar, we need to prevent the learning algorithm from inserting an arbitrary number of non-terminal nodes in the chain of generations. "Requiring some text to be generated at each [production] step is enough for inference to remain tractable." (Malik et al., 2021)

These observations may be closely related. Adding direct connections from the data to every latent activation function in the network removes an enormous number of degenerate states from the solution space.

The fact that the add & normalize is the layer in the vanilla transformer that has residual connections to the data tells us that this is the layer that corresponds to the $\wedge$ in our generative production. In appendix D we expand on this correspondence.

### A.2    CONNECTING THE ENCODER TO THE DECODER

When performing marginalization of a probability distribution represented by a directed acyclic graph, it has long been known that marginalization (probabilistic message passing) converges after a single forward pass and backward pass through the graph (Pearl, 1988; MacKay, 2003).

The forward pass in our generative model corresponds to the encoder, and the backward pass to the decoder. Carefully tracing the forward and backward marginalizations (as we do in appendix B on activation functions below), shows us that the inductive transformer could potentially benefit from a slightly different connectivity between the encoder and decoder compared to a vanilla transformer. Vanilla transformers likely learn to work around this limitation in the connectivity by propagating the same information through the residual stack.

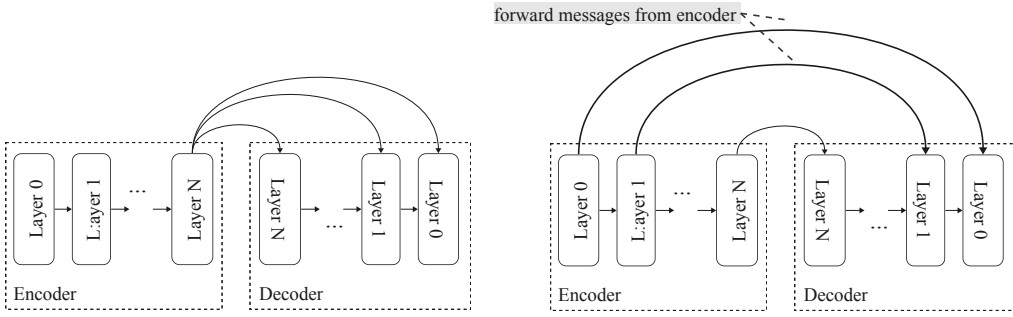

Figure 3: How the Connectivity from Encoder to Decoder Differs in the Inductive Transformer.

# B ACTIVATION FUNCTIONS

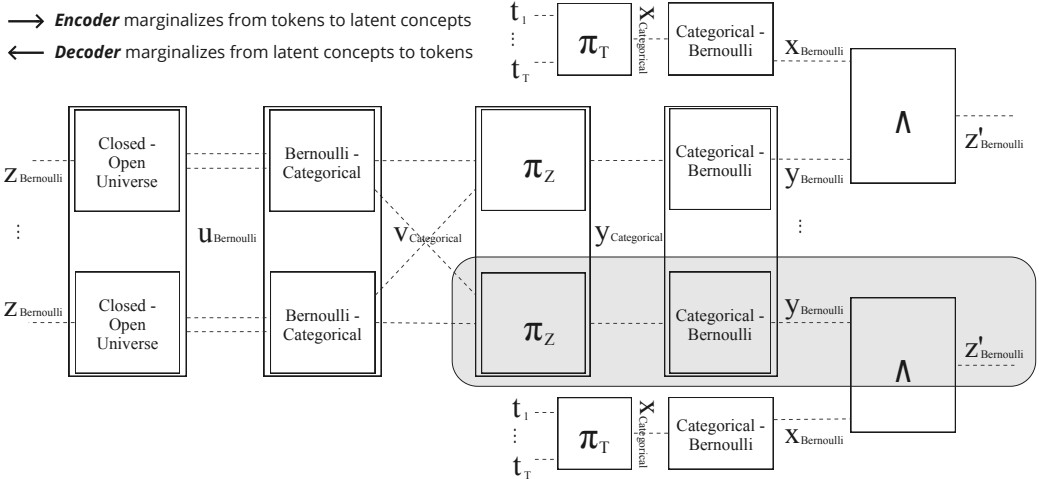

Figure 4: Factor Graph For Inductive Transformer

The sequence of marginalization operations previously mentioned in section 2 correspond to layers in the inductive transformer. See figure 4 for a detailed review of a single layer. The *encoder* layers from the input towards the decoder are,

1. Closed-Open Universe

2. Bernoulli-Categorical

3. Attention $\pi$ and Token $\pi$

4. Categorical-Bernoulli

5. $\wedge$

6. Repeat.

The *decoder* layers from the encoder towards the output are,

1. $\wedge$

2. Bernoulli-Categorical

3. Attention $\pi$ and Token $\pi$

4. Categorical-Bernoulli

5. Open-Closed Universe

6. Repeat.

We detail each layer below.

### B.1 OPEN CLOSED UNIVERSE: REPRESENTING AN OPEN UNIVERSE NON-PARAMETRIC MODEL IN A CLOSED UNIVERSE NEURAL NETWORK WITH FIXED LAYER WIDTH

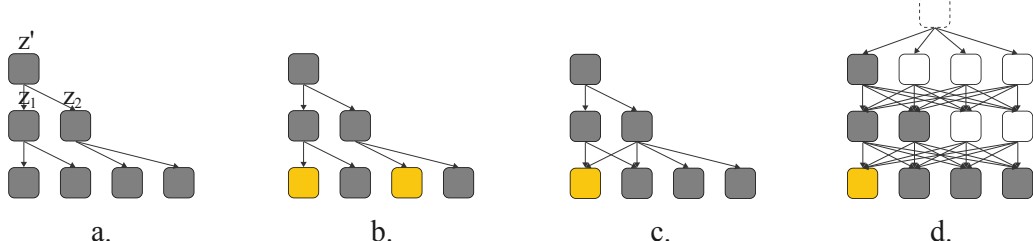

Figure 5: One grey box in this figure corresponds to the grey box in figure 1 (a.) Open Universe Model: Every child node has a single parent node. Parents make categorical choices over existing children. A child that is activated has only one parent activating it. (b.) Open Universe Model: If two parent nodes would like to activate the same child, we must make a copy of that child. The two yellow rectangles represent the same concept. The open universe model must duplicate it in order to allow two parents to utilize it. (c.) Closed Universe Model: Duplicate concepts are merged into a single node. A child, therefore, can have more than one parent node. Since we do not have a potentially infinite layer width, this allows for re-use of limited closed universe resources, and allows for use of scalable closed-universe solvers such as back-propagation. To do the math correctly when a single child is simultaneously selected by more than one parent, we must use a conditional distribution where if one or more parents selects the child, then the child is activated. (d.) Closed Universe Model: We can make every layer the same width in order to allow for more routes through the model to explain away the data. This is equivalent to an open universe grammar with N children under the root, and then only N different types of grandchild node, and so forth.

| child | $\text{parent}_0$ | $\text{parent}_1$ | $p(\text{parent}_0|\text{child}, \text{parent}_1)$ |
|-------|---------|---------|--------------------|
| 1 | 1 | 0 | 1/4 |
| 1 | 0 | 1 | 1/4 |
| 1 | 1 | 1 | 1/4 |
| 0 | 0 | 0 | 1/4 |
| 1 | 0 | 0 | 0 |
| 0 | 1 | 0 | 0 |
| 0 | 0 | 1 | 0 |
| 0 | 1 | 1 | 0 |

Table 4: This table represents the states of an Open-Closed-Universe factor with two parents and one child. In the decoder, we will need the conditional probability of the child given all of the parents, $p(\text{child}|\text{parent}_0, \text{parent}_1)$. In the encoder, we will need the conditional probability of one of the parents, given the child and the other parent, $p(\text{parent}_0|\text{child}, \text{parent}_1)$ and $p(\text{parent}_1|\text{child}, \text{parent}_0)$. All of these conditional distributions can be derived from this table. In an open-universe model, there would be two copies of the child, so that each parent has its own unique children, and therefore each child would have only one parent. When we combine multiple children into a single node with multiple parents, then that combined child should be active if one or more of its parents are active. This is what we do when we represent an open universe within a closed universe model. We represent allowed states with a probability of $1/4$. Disallowed states have a probability of $0$. probability $= 1$ states have the child $= 1$ when any of its parents are a $1$ or have the child $= 0$ when all parents are zeros. Disallowed states (rows) violate those rules and involve one or more parents being a $1$ while the child $= 0$ or the child being a $1$ when both parents are $0$.

An open universe model does not have a fixed layer width. It can sample new branches and nodes into existence when they are needed in order to explain away the data with greater likelihood. In a "closed universe" the network has fixed width and depth. When we use back-propagation as the solver in such models, it optimizes the weights between existing nodes but it does not create new nodes nor new edges. Therefore a parent node cannot gain new children, we can only increase the weight between the parent and preexisting child nodes. In principle, we might think that there is no real difference between creating a new node and connecting to an existing node that has not yet been utilized. But in practice, the form of the activation function of the child node must be adjusted to allow for having multiple parents.

It is easiest to first consider the *decoder* open-closed-universe-connector where we marginalize for the child given the parents. We assume all distributions are Bernoulli,

$$p(\text{child}) = \sum_{\text{parent}_0} \sum_{\text{parent}_1} p(\text{child}|\text{parent}_0, \text{parent}_1)p(\text{parent}_0)p(\text{parent}_1), \tag{6}$$

where

$$p(\text{child}|\text{parent}_0, \text{parent}_1) = \begin{cases} 1 \text{ when } \sum_i \text{parent}_i \geq 1 \\ 0 \text{ when } \sum_i \text{parent}_i = 0. \end{cases} \tag{7}$$

We often refer to the result on the left-hand side of this equation as a "message". A message is the output being computed by a marginalization operation. An outgoing message in the decoder, we multiply the conditional probability derived from this table by incoming messages $p(\text{parent}_0)$ and $p(\text{parent}_1)$. The message is therefore,

$$\begin{aligned} p(\text{child} = 0) &= p(\text{parent}_0 = 0)p(\text{parent}_1 = 0) \\ p(\text{child} = 1) &= p(\text{parent}_0 = 1)p(\text{parent}_1 = 1) \\ &+ p(\text{parent}_0 = 0)p(\text{parent}_1 = 1) \\ &+ p(\text{parent}_0 = 1)p(\text{parent}_1 = 0). \end{aligned} \tag{8}$$

Finally, we will normalize the outgoing message so that $p(\text{child} = 1) + p(\text{child} = 0) = 1$.

For the *encoder* closed-to-open universe factor we instead marginalize for a parent, given the child and other parents:

$$p(\text{parent}_0) = \sum_{\text{child}} \sum_{\text{parent}_1} p(\text{parent}_0|\text{child}, \text{parent}_1)p(\text{child})p(\text{parent}_1) \tag{9}$$

and,

$$p(\text{parent}_1) = \sum_{\text{child}} \sum_{\text{parent}_0} p(\text{parent}_1|\text{child}, \text{parent}_0)p(\text{child})p(\text{parent}_0) \tag{10}$$

Table 4 also defines the distribution $p(\text{parent}_0|\text{child}, \text{parent}_1)$ yielding,

$$\begin{aligned} p(\text{parent}_0 = 1) &= p(\text{child} = 1)p(\text{parent}_1 = 0) \\ &+ p(\text{child} = 1)p(\text{parent}_1 = 1) \\ p(\text{parent}_0 = 0) &= p(\text{child} = 1)p(\text{parent}_1 = 1) \\ &+ p(\text{child} = 0)p(\text{parent}_1 = 0). \end{aligned} \tag{11}$$

Note that we do inference in the encoder first, before we do inference in the decoder. This means encoder inference proceeds without information from decoder inference. And that means that on the

right hand side of equation 11, $p(\text{parent}_1 = 1)$ and $p(\text{parent}_1 = 0)$ will both be $0.5$. And *that* means we will always have,

$$p(\text{parent}_0 = 1) = p(\text{child} = 1)$$
$$p(\text{parent}_0 = 0) = 0.5. \tag{12}$$

In other words, for the encoder, the closed-to-open universe gate acts like a pass-through:

$$p(\text{parent}_i = 1) = p(\text{child} = 1) \tag{13}$$

### B.2 BERNOULLI-CATEGORICAL

As we saw above, when we treat certain variables in our model as a joint Categorical rather than as a collection of independent Bernoulli variables, we reduce the computational complexity from exponential to polynomial. In other parts of our model, however, we will find that we require the independent Bernoulli variables. We are therefore compelled to transform between the two representations.

Consider the conditional distribution involving two Bernoulli variables and one categorical variable, $p(y_{\text{Categorical}} = i | y_{\text{Ber}}^j, y_{\text{Ber}}^k)$ where $j = \text{dog}$ and $k = \text{cat}$, and $i$ varies over the states dog and cat. This is represented in table 5.

| $y_{\textbf{Categorical}}$ | $y_{\textbf{Ber}}^{\textbf{dog}}$ | $y_{\textbf{Ber}}^{\textbf{cat}}$ | $p(y_{\text{Categorical}}|y_{\text{Ber}}^{\text{dog}}, y_{\text{Ber}}^{\text{cat}})$ |
|:---:|:---:|:---:|:---:|
| dog | 1 | 0 | 1/2 |
| cat | 0 | 1 | 1/2 |
| dog | 1 | 1 | 0 |
| dog | 0 | 0 | 0 |

Table 5: Each row in this table represents the probability of a conditional distribution for a state of the Categorical variable given two Bernoulli variables. In this example, the state of the Categorical can be either dog or cat. One Bernoulli is providing information about if there is a dog or not. A second Bernoulli tells us if we see a cat or not. The conditional distribution of the Categorical given both Bernoullis is $p(y_{\text{Categorical}} \in \{\text{dog}, \text{cat}\}|y_{\text{Ber}}^{\text{dog}} \in \{0, 1\}, y_{\text{Ber}}^{\text{cat}} \in \{0, 1\})$. When the incoming dog-Bernoulli is a 1 and the incoming cat-Bernoulli is a 0, and the categorical is set to its dog state, then the conditional probability is $1/2$. Similarly we have a probability=$1/2$ in the second row where the cat-Bernoulli is 1, the dog-Bernoulli is 0, and the categorical is set to the cat state. However, as we see in the third and fourth rows, when the dog-Bernoulli and the cat-Bernoulli are both ones or both zeros, these are probability=0 states. The Bernoulli-categorical factor obeys the constraint that dog and cat can't both be 1, nor can they both be 0. Therefore the probabilities in the last two rows are 0.

Using the conditional distribution above, we can marginalize out the Bernoulli variables,

$$p(y_{\text{Categorical}}) = \sum_{y_{\text{Ber}}^j, y_{\text{Ber}}^k} p(y_{\text{Categorical}}|y_{\text{Ber}}^j, y_{\text{Ber}}^k)p(y_{\text{Ber}}^j)p(y_{\text{Ber}}^k), \tag{14}$$

$$\tag{15}$$

yielding,

$$p(y_{\text{Categorical}} = \text{dog}) = p(y_{\text{Ber}}^{\text{dog}} = 1)p(y_{\text{Ber}}^{\text{cat}} = 0)/\text{norm}$$
$$p(y_{\text{Categorical}} = \text{cat}) = p(y_{\text{Ber}}^{\text{dog}} = 0)p(y_{\text{Ber}}^{\text{cat}} = 1)/\text{norm}, \tag{16}$$

where we normalize by dividing by norm $= p(y_{\text{Categorical}} = \text{dog}) + p(y_{\text{Categorical}} = \text{cat})$.

The conditional distribution is once again given by the truth table 5 and reads:

$$p(y_{\text{Categorical}} = i | y_{\text{Ber}}^j = 1, y_{\text{Ber}}^k = 0) = \begin{cases} 1, & \text{if } i = j \text{ and } i \neq k \\ 0, & \text{otherwise} \end{cases} \quad . \tag{17}$$

We illustrate in an example with three states, {dog, cat, bird},

$$\begin{aligned} p(\text{dog}) &= p_{\text{dog}}(1)p_{\text{cat}}(0)p_{\text{bird}}(0) \\ p(\text{cat}) &= p_{\text{dog}}(0)p_{\text{cat}}(1)p_{\text{bird}}(0) \\ p(\text{bird}) &= p_{\text{dog}}(0)p_{\text{cat}}(0)p_{\text{bird}}(1). \end{aligned} \tag{18}$$

Using the fact that $p_{\text{dog}}(0) = 1 - p_{\text{dog}}(1)$ we can rewrite $p(\text{dog})$ as,

$$\begin{aligned} p(\text{dog}) &= (1 - p_{\text{dog}}(0))p_{\text{cat}}(0)p_{\text{bird}}(0) \tag{19} \\ p(\text{dog}) &= p_{\text{cat}}(0)p_{\text{bird}}(0) - p_{\text{dog}}(0)p_{\text{cat}}(0)p_{\text{bird}}(0). \tag{20} \end{aligned}$$

We could rewrite $p(\text{cat})$ and $p(\text{bird})$ similarly. Since we will normalize the Categorical at the end, we can divide $p(\text{dog})$, $p(\text{cat})$ and $p(\text{bird})$ by the same constant at any time during the derivation. We divide by $p_{\text{dog}}(0)p_{\text{cat}}(0)p_{\text{bird}}(0)$, which yields

$$\begin{aligned} p(\text{dog}) &= \frac{1}{p_{\text{dog}}(0)} - 1 = \frac{p_{\text{dog}}(1)}{p_{\text{dog}}(0)} \\ p(\text{cat}) &= \frac{1}{p_{\text{cat}}(0)} - 1 = \frac{p_{\text{cat}}(1)}{p_{\text{cat}}(0)} \\ p(\text{bird}) &= \frac{1}{p_{\text{bird}}(0)} - 1 = \frac{p_{\text{bird}}(1)}{p_{\text{bird}}(0)} \end{aligned} \tag{21}$$

This distribution will need to be normalized. But we can already see from the previous section that, because $p_i(0)$ is constant the encoder remains a simple pass-through:

$$p(\text{parent}_i) = p(\text{child} = 1) \tag{22}$$

In the decoder, we also convert from Bernoulli to Categorical as we marginalize from the $\wedge$ to the $\pi_z$'s and $\pi_T$'s. The marginal for the Categorical variable is computed using decoder Bernoulli variables. There is no need for information from the encoder.

### B.3 ENCODER $\pi$

$\pi_T$ is used as a choice over tokens, $\pi_Z$ as a choice over concepts in the layer below, and $\pi_\rho$ as a choice over positions.

Let $z_i$ be a Bernoulli variable indicating whether or not the $i$-th $\wedge_i$ in the encoder layer below is a 1. Let $y$ be a Bernoulli that is a 1 when one and only one of these $z_i$ is a 1. Starting with the joint distribution $p(y, z_1 \ldots, z_Z)$, in the encoder we would like to marginalize to find $p(y)$,

$$\begin{aligned} p(y) &= \sum_{z_1 \in \{0,1\}} \cdots \sum_{z_Z \in \{0,1\}} p(y, z_1, \ldots, z_Z) \\ &= \sum_{z_1 \in \{0,1\}} \cdots \sum_{z_Z \in \{0,1\}} p(y|z_1, \ldots, z_Z)p(z_1) \ldots p(z_Z) \end{aligned} \tag{23}$$

| $y_{\text{categorical}}$ | $\mathbf{z}_1$ | $\mathbf{z}_2$ | probability |
|:---:|:---:|:---:|:---:|
| 1 | 1 | 0 | 1/4 |
| 1 | 0 | 1 | 1/4 |
| 0 | 1 | 1 | 1/4 |
| 0 | 0 | 0 | 1/4 |
| 0 | 0 | 1 | 0 |
| 1 | 1 | 1 | 0 |

Table 6: $\pi$ is an "exactly-one" constraint such that $y = 1$ if and only if $\sum_i z_i = 1$. In other words, only a single $z_i$ can be a 1 when $y$ is 1.

The probability values in table 6 can be expressed as

$$p(y = 1|z_i, \ldots, z_Z) = \begin{cases} 1 \text{ when } \sum_{i \in \{1, \ldots, Z\}} z_i = 1 \\ 0 \text{ when } \sum_{i \in \{1, \ldots, Z\}} z_i > 1 \text{ or when } \sum_{i \in \{1, \ldots, Z\}} z_i = 0, \end{cases} \tag{24}$$

where the last step is to normalize the distribution.

For example, with only two inputs $p(z_1)$ and $p(z_2)$ we have,

$$p(y = 1) = p(z_1 = 1)p(z_2 = 0) + p(z_1 = 0)p(z_2 = 1)$$
$$p(y = 0) = p(z_1 = 0)p(z_2 = 0) + p(z_1 = 1)p(z_2 = 1). \tag{25}$$

As we can see, when the $z_i$ are Bernoulli variables, calculating $p(y)$ becomes $O(2^Z)$, where $Z$ is the number of $z_i$'s. If, however, we treat the $z_i$'s as states of a single categorical rather than as independent Bernoullis, we will see that the computational complexity can be simplified significantly to $O(Z)$. In order to do this we must treat the entire layer of $z_i$'s as a single joint variable, $z$.

$$p(y) = \sum_i p(y, z) \tag{26}$$

$$= \sum_i p(y|z)p(z) \tag{27}$$

$$= \sum_i \omega_i p(z_i), \tag{28}$$

where $\omega$'s are learned weights and $i$ indexes the $\wedge_i$'s in the layer below that are sending the output from their marginalization to the inputs of this encoder $\pi$. This inner product calculation has complexity $O(Z)$, and we need to perform this calculation across the entire layer width $Z$, leading to an overall complexity that is $O(Z^2)$ versus the Bernoulli case where the complexity was $O(2^Z)$.

The word or token $\pi_T$ is very similar. With token vocabulary size $T$ and layer width of size $Z$ (indexed by $l$), we have a calculation of complexity $O(Z \cdot T)$,

$$p(x_l) = \sum_{\tau \in (1, \ldots, T)} \omega_{l, \tau} p(t_\tau). \tag{29}$$

The token $\pi$, instead of connecting to the layer below, connects directly to the words/tokens in the input data. In a probabilistic generative grammar this is referred to as a "terminal" node, in contrast to a non-terminal which generates another layer of the model. In deep learning models, the connection from a terminal node to the data is referred to as a residual or skip connection.

In both cases (token and attention), we normalize over the layer before providing it to the next layer in the network, since we treat the entire layer of $y$ edges as a single joint distribution, $p(y)$.

In summary, compared to the Bernoulli version that we derived first, it makes sense to use this categorical version of $\pi$, because it simplifies the computational complexity from $O(2^n)$ to $O(n^2)$. Furthermore, in the decoder if we used a Bernoulli version, when computing a particular $p(z_i)$ for exact forward-backward marginalization the decoder marginalization would be required to utilize the information from the encoder, $p(z_{j\neq i})$. With $p(z)$ being a single joint variable, however, there are two incident edges to a decoder $\pi$, $p(x)$ and $p(z)$. This means when computing the marginalization to find $p(z)$ in the decoder, we only need the decoder message $p(x)$. We do not need any messages provided by the encoder.

## B.4    CATEGORICAL-BERNOULLI

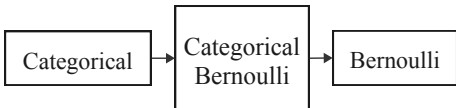

Figure 6: Factor Graph for Categorical to Bernoullis Factor

Let us start with an example with only one Bernoulli (a dog-Bernoulli $\in \{0, 1\}$) and one Categorical (with states $\in \{\text{dog, cat}\}$, see figure 6).

$$
\begin{aligned}
p(z_{\text{Ber}}^{\text{dog}} = 1) &= \sum_{z_{\text{Categorical}} \in \text{dog, cat}} p(z_{\text{Ber}}^{\text{dog}} = 1, z_{\text{Categorical}}) \\
&= \sum_{z_{\text{Categorical}} \in \text{dog, cat}} p(z_{\text{Ber}}^{\text{dog}} = 1 | z_{\text{Categorical}}) p(z_{\text{Categorical}}) \\
&= p(z_{\text{Ber}}^{\text{dog}} = 1 | z_{\text{Categorical}} = \text{dog}) p(z_{\text{Categorical}} = \text{dog}) \\
&\quad + p(z_{\text{Ber}}^{\text{dog}} = 1 | z_{\text{Categorical}} = \text{cat}) p(z_{\text{Categorical}} = \text{cat}) \\
&= 1 \cdot p(z_{\text{Categorical}} = \text{dog}) + 0 \cdot p(z_{\text{Categorical}} = \text{cat}) \\
&= p(z_{\text{Categorical}} = \text{dog})
\end{aligned}
\tag{30, 31}
$$

$$
\begin{aligned}
p(z_{\text{Ber}}^{\text{dog}} = 0) &= p(z_{\text{Ber}}^{\text{dog}} = 0 | z_{\text{Categorical}} = \text{dog}) p(z_{\text{Categorical}} = \text{dog}) \\
&\quad + p(z_{\text{Ber}}^{\text{dog}} = 0 | z_{\text{Categorical}} = \text{cat}) p(z_{\text{Categorical}} = \text{cat}) \\
&= p(z_{\text{Categorical}} = \text{cat}).
\end{aligned}
\tag{32, 33}
$$

The Bernoulli and the Categorical in this example are really saying the same thing. The mutually exclusive alternative to a dog in the Categorical is a cat. The mutually exclusive alternative to a 1 for the dog-Bernoulli is a 0. The probability of cat in the Categorical should be equal to the probability of a 0 in the Bernoulli.

Now let's introduce an explicit cat-Bernoulli into the example.

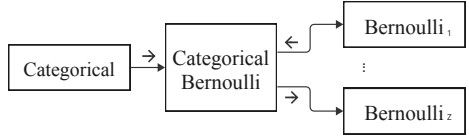

Figure 7: Factor Graph for Categorical to Bernoulli Factor. The arrows indicate the direction of the messages.

Recall that the rule for marginalization within a factor graph is that we compute an outgoing "message" from a factor using the incoming "messages" on all other edges incident to the factor (Yedidia et al., 2003). This is not important in the encoder when we are still doing a forward marginalization pass and there is no useful "backward" information available from the decoder yet. In the decoder, however, we must consider "forward" information from the encoder if we wish to do exact inference to exactly infer all of the latent variables given all of the available data.

As we can see in figure 7, when we add a Bernoulli for cat, and compute an outgoing message for the dog-Bernoulli, we also need an incoming message from the cat-Bernoulli.

We recall table 5, but move the columns around to fit our current situation. Each row provides the conditional probability, $p(y_{\text{Ber}}^{\text{dog}}|y_{\text{Categorical}}, y_{\text{Ber}}^{\text{cat}})$, in a situation with two Bernoulli variables, one for whether dog is True or False (1 or 0 respectively), and similarly one for cat, as well as one Categorical with two states ("dog" and "cat").

| $z_{\textbf{Ber}}^{\textbf{dog}}$ | $z_{\textbf{Categorical}}$ | $z_{\textbf{Ber}}^{\textbf{cat}}$ | $p(y_{\text{Ber}}^{\text{dog}}|y_{\text{Categorical}}, y_{\text{Ber}}^{\text{cat}})$ |
|---|---|---|---|
| 1 | dog | 0 | 1/2 |
| 0 | cat | 1 | 1/2 |
| 1 | dog | 1 | 0 |
| 0 | dog | 0 | 0 |

Table 7:

We care about computing the marginal probability for a given Bernoulli, e.g. $p_{\text{Ber}}^{\text{dog}}(1)$,

$$p_{\text{Ber}}^{\text{dog}}(1) = \sum_{z_{\text{Categorical}}} \sum_{z_{\text{Ber}}^{\text{cat}}} p(z_{\text{Ber}}^{\text{dog}}, z_{\text{Ber}}^{\text{cat}}, z_{\text{Categorical}})$$

$$= \sum_{z_{\text{Categorical}}} \sum_{z_{\text{Ber}}^{\text{cat}}} p(z_{\text{Ber}}^{\text{dog}}|z_{\text{Ber}}^{\text{cat}}, z_{\text{Categorical}}) p(z_{\text{Ber}}^{\text{cat}}) p(z_{\text{Categorical}}). \tag{34}$$

For this calculation, we need the conditional probability $p(z_{\text{Ber}}^{\text{dog}} = 1|\ldots)$,

$$p(z_{\text{Ber}}^{\text{dog}} = 1|z_{\text{Categorical}} = j, z_{\text{Ber}}^{\text{cat}}) = \begin{cases} 1, & \text{when } z_{\text{Categorical}} = \text{dog}, z_{\text{Ber}}^{\text{cat}} = 0 \\ 0, & \text{otherwise} \end{cases}, \tag{35}$$

where, as always, the distribution must be subsequently normalized. We can write this more generally, with $i$ instead of "dog" and $j$ for the state of $z_{\text{Categorical}} = j$, and with $Z$ Bernoulli variables indexed by $k$,

$$p(z_{\text{Ber}}^i = 1 | z_{\text{Categorical}} = j, z_{\text{Ber}}^1, \ldots, z_{\text{Ber}}^{i-1}, z_{\text{Ber}}^{i+1}, \ldots, z_{\text{Ber}}^Z) \tag{36}$$

$$= \begin{cases} 1 \text{ when } j = i, \text{ and } z_{\text{Ber}}^k = 0 \; \forall k \neq i \\ 0, \text{ otherwise} \end{cases} . \tag{37}$$

To compute the marginal probability $p_{\text{Ber}}^{\text{dog}}(0)$, we will need the conditional probability $p(z_{\text{Ber}}^{\text{dog}} = 0 | \ldots)$. This conditional probability is a 1 when $z_{\text{Ber}}^{\text{cat}} = 1$ and $z_{\text{Categorical}} = \text{cat}$. In other words, we need an allowed state with $z_{\text{Ber}}^{\text{dog}} = 0$, which means that overall we need a joint "cat" state.

More generally, if we have more Bernoulli variables, for example a "bird" variable, then an allowed state always has $z_{\text{Categorical}} = j$ and also has $z_{\text{Ber}}^j = 1$ and all other Bernoullis $z_{\text{Ber}}^{i \neq j} = 0$. Therefore the $\sum_i z_{\text{Ber}}^i = 1$. Putting this all together we write,

$$p(z_{\text{Ber}}^i = 0 | z_{\text{Categorical}} = j, z_{\text{Ber}}^1, \ldots, z_{\text{Ber}}^{k-1}, z_{\text{Ber}}^{k=j}, z_{\text{Ber}}^{k+1}, \ldots, z_{\text{Ber}}^Z) \tag{38}$$

$$= \begin{cases} 1 \text{ when } \sum_k z_{\text{Ber}}^k = 1 \text{ and } j \neq i \\ 0 \text{ otherwise} \end{cases} . \tag{39}$$

The computation for $p(z_{\text{Ber}}^i = 0)$ will contain only one product per Bernoulli. So the computations involving incoming Bernoulli messages are not prohibitively complex. We do not need to compute them at all, however. In the encoder we use a categorical-Bernoulli as we send forward messages from the $\pi$ to the $\wedge$ and there is not yet any backward information to incorporate into our calculation.

In the decoder, we compute the Categorical-Bernoulli as we send marginals from the $\pi$ to the open universe. If in we choose to ignore the forward messages from the encoder, we can simplify our computation considerably. We can set the encoder marginals to be non-informative for the decoder Categorical-Bernoulli, $p(z_{\text{Ber}}) = 0.5$. Let us do a pedagogical example with a categorical and three Bernoulli variables representing "dog", "cat", and "bird".

$$p(z_{\text{Ber}}^{\text{dog}} = 1) = p(z_{\text{Categorical}} = \text{dog})p(z_{\text{Ber}}^{\text{cat}} = 0)p(z_{\text{Ber}}^{\text{bird}} = 0), \tag{40}$$

$$p(z_{\text{Ber}}^{\text{dog}} = 0) = p(z_{\text{Categorical}} = \text{cat})p(z_{\text{Ber}}^{\text{cat}} = 1)p(z_{\text{Ber}}^{\text{bird}} = 0),$$

$$+ p(z_{\text{Categorical}} = \text{bird})p(z_{\text{Ber}}^{\text{cat}} = 0)p(z_{\text{Ber}}^{\text{bird}} = 1) \tag{41}$$

Now if $p(z_{\text{Ber}}^{\text{bird}} = 1) = p(z_{\text{Ber}}^{\text{bird}} = 0) = p(z_{\text{Ber}}^{\text{cat}} = 1) = p(z_{\text{Ber}}^{\text{cat}} = 0) = 0.5$ then,

$$p(z_{\text{Ber}}^{\text{dog}} = 1) = p(z_{\text{Categorical}} = \text{dog})(0.25), \tag{42}$$

$$p(z_{\text{Ber}}^{\text{dog}} = 0) = p(z_{\text{Categorical}} = \text{cat})(0.25) + p(z_{\text{Categorical}} = \text{bird})(0.25) \tag{43}$$

More generally, after normalizing this becomes,

$$p(z_{\text{Ber}}^i = 1) = p(z_{\text{Categorical}} = i)$$

$$p(z_{\text{Ber}}^i = 0) = \sum_{j \neq i} p(z_{\text{Categorical}} = j) \tag{44}$$

### B.5 ENCODER $\wedge$

The encoder $\wedge$ imposes the constraint z = AND(x,y), where $x$, $y$, and $z$ are random binary variables. In the truth table below, each row lists a probability for a joint state on $x$, $y$, and $z$. The (four) allowed states have non-zero probabilities normalized to $1/4$. The disallowed states have zero probability.

| x | y | z | $p(z|x,y)$ |
|---|---|---|---|
| 0 | 0 | 0 | 1/4 |
| 0 | 0 | 1 | 0 |
| 0 | 1 | 0 | 1/4 |
| 0 | 1 | 1 | 0 |
| 1 | 0 | 0 | 1/4 |
| 1 | 0 | 1 | 0 |
| 1 | 1 | 0 | 0 |
| 1 | 1 | 1 | 1/4 |

One way to think about these modules is to imagine that $x$ is a stochastic stream of 0's and 1's, where the 1's occur with probability $p_x$ and 0's with probability $1 - p_x$. Similarly, imagine $y$ as a stochastic stream of 0's and 1's, where the 1's occur with probability $p_y$ and 0's with probability $1 - p_y$. When we marginalize to find $p(z = 1)$, we are answering the question, "if the stochastic streams for $x$ and $y$ passed through an ordinary AND logic gate, what would be the percentage of 1's in the output stream, $z$?"

We answer this question by computing the marginal probability for $p(z)$,

$$
\begin{aligned}
p(z) &= \sum_{x,y} p(x, y, z) \\
&= \sum_{x,y} p(z|x, y)p(x)p(y) \\
&= \sum_{x,y} \delta(z - \wedge(x, y))p(x)p(y) \\
p(z_l) &= \sum_{x,y} \delta(z - \wedge(x_l, y_l))p(x_l)p(y_l)
\end{aligned}
\tag{45}
$$

where $l$ indexes across a layer of the model, and $p(z)$ will be normalized as a final step in the calculation. Expanding the sums and products subject to the constraint, we have,

$$
\begin{aligned}
p(z_l = 1) &= p(x_l = 1)p(y_l = 1) \\
p(z_l = 0) &= p(x_l = 0)p(y_l = 0) \\
&\quad + p(x_l = 0)p(y_l = 1) \\
&\quad + p(x_l = 1)p(y_l = 0).
\end{aligned}
\tag{46}
$$

## B.6 DECODER $\wedge$

The decoder $\wedge$ is very similar to the encoder $\wedge$,

$$
\begin{aligned}
p_{\text{y decode}}(1) &= p_{\text{x encode}}(0)p_{\text{z decode}}(0) + p_{\text{x encode}}(1)p_{\text{z decode}}(1) \\
p_{\text{y decode}}(0) &= p_{\text{x encode}}(0)p_{\text{z decode}}(0) + p_{\text{x encode}}(1)p_{\text{z decode}}(0)
\end{aligned}
\tag{47}
$$

As we can see, when the decoder $\wedge$ is computing $p_{\text{decode}}(y)$, it should not only use $p_{\text{decode}}(z)$, it should also use $p_{\text{encode}}(x)$. Similarly, when the decoder $\wedge$ is computing $p_{\text{decode}}(x)$, marginalization should use $p_{\text{encode}}(y)$. However, this would mean that to predict $p_{\text{decode}}(x)$ at a given position, the model is using information from tokens to the right in the text. Indeed, not using $p_{\text{encode}}(y)$ in the

decoder is the equivalent of forward-only message passing in belief propagation (Vigoda, 2003; Murphy, 2002) which predicts the next symbol in a sequence based only on previous signals. Thus, we could choose not to use $p_{\text{encode}}(x)$ and $p_{\text{encode}}(y)$ in the decoder to train it for this task.

The next layer in the decoder is the Bernoulli-Categorical, which we have already described above. Moving on, we describe the decoder $\pi_T$.

### B.7 DECODER TOKEN $\pi$

Let us consider a single $\pi_T$ within a layer of the model. $p(x)$ is an output of the decoder $\wedge$ which becomes the input to this $\pi_T$. The marginal probability of generating token $t$ from $\pi_T$ is,

$$
\begin{aligned}
p(t = \tau) &= p(t = \tau | x) p(x), \\
p(t = \tau) &= \omega_\tau \cdot p(x),
\end{aligned}
\tag{48}
$$

Sampling from this distribution, we would get a single token with probability $\omega_t$, where $\omega$'s are learned weights.

We are treating $p(t)$ as a Categorical with a state for each token, so we might think that we need to normalize such that $\sum_\tau p(t_\tau) = 1$. This would be incorrect. We do use normalized (learned) weights such that, $\sum_\tau \omega_\tau = 1$, but we do not normalize the output of the decoder $\pi_T$'s. The reason is that this is another situation where we must take into account the open-closed universe factor. Each $\pi_t^l$ in the layer indexed by $l$ has some probability of outputting a given token, e.g. "cat". These $\pi_t^l$'s share children, and all of their probabilities for outputting a given token must be summed.

We can see in figure 4, that the output of the decoder $\pi_Z$ goes through a Categorical-to-Bernoulli and then into an open-closed universe layer, both as described above in appendices B.1 and B.4.

What we are really doing when we do not normalize the outputs of an individual decoder $\pi_T$ or $\pi_Z$ within a single layer, is allowing the $\pi$'s to each participate proportionally in activating their shared children and have the normalization come out correctly as it would in an open universe.

For example, for a particular $\pi_t$, when the input is p(x=1) = 80% and p(x=0)=20%, and the $\pi_t$ weights are dog weight = 50% and cat weight = 50%, then the probability of outputting the word dog, p(dog)=40%, p(cat)=40% and the (un-normalized) probability of outputting no token at all is p(no output)=20%. When outputting *no output*, this $\pi_T$ is allowing some other $\pi_{j \neq i}$ to generate the token at the given location in the data.

## C SELF-ATTENTION IN THE INDUCTIVE TRANSFORMER

### C.1 VANILLA SELF-ATTENTION

Many variants of the transformer have appeared since the publication of the original "vanilla" version, so we should not be overly pedantic as we derive the corresponding inductive transformer attention mechanism. Our derivation will follow more closely the "relative positional encoding" formulation of Transformer-XL model than the vanilla version (Dai et al., 2019).

Regardless, it is useful to recall that the core of self-attention in a vanilla transformer (Vaswani et al., 2017) is,[3]

$$
Y_i = \sum_j \text{softmax}\left(\frac{Q_i \cdot K_j^T}{\sqrt{d}}\right) V_j,
\tag{49}
$$

where $Q$, $K$, and $V$ are the query, key, and value matrices, each formed by applying matrices $W^Q$, $W^K$, $W^V$ to the embedding vector input to this attention layer at each position, and where $d$ is

---

[3]We suppress chunking which is used for parallel computing in hardware, but is a distraction when describing the model.

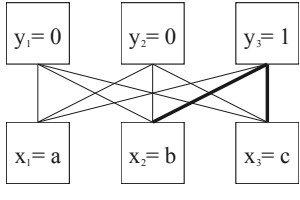
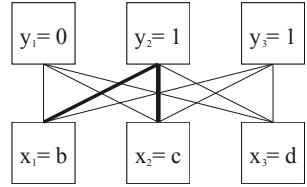

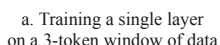

a. Training a single layer
on a 3-token window of data

b. Training with the data shifted to the
left by one token

Figure 8: We are training a single layer on a 3-token window of data. In (a) we have the first 3-token window of data. In (b) the data is shifted to the left by one token, so that a new token 'd' enters the window from the right, and a new target output '1' also enters from the right.

the embedding dimension. Because of the softmax operation, each row of the $Q_i \cdot K_j^T$ matrix is a normalized set of weights for linearly combining vectors $V_j$. Note that the $Y$ here is from the notation in the original vanilla transformer paper, and is not identical to the $Y$ in figure 4.

## C.2 THE GOAL OF THE SELF-ATTENTION MECHANISM

In figure 8 we have three input variables at three positions $\{x_1, x_2, x_3\}$, and three output variables at three positions $\{y_1, y_2, y_3\}$ forming a standard single-layer perceptron.

When we set the input $\{x_1 = a, x_2 = b, x_3 = c\}$, we want the output to be $\{y_1 = 0, y_2 = 0, y_3 = 1\}$, so the network needs to learn that $x_2 = b$ implies $y_3 = 1$, and $x_2 = c$ may imply $y_3 = 0$. Once we shift the data to the left by one position, however, we now need to learn the exact opposite, that $x_2 = b$ implies $y_3 = 0$, and $x_2 = c$ may imply $y_3 = 1$. By training with the data shifted by one position, we have begun to erase what we had previously learned.

The problem is that the network is trying to learn functions of tokens at absolute positions in the data, when we really want the network to learn relationships involving the relative positions of tokens. This is the essential function of the attention mechanism.

As the data traverses the window but remains in view, we want it to continue to reinforce the same relationships in the model. As novel data enters the window, we want it to add new relationships in the model. We also want this to be the case not just in a single input layer, but also in the deeper layers of the network.

Our goal is therefore to augment or modify our production so that it is invariant to shifts in the absolute positions of tokens, and instead considers tokens at relative positions.

Instead of describing matrix operations on embedding vectors to compute $Y$ as we did in equation 49, we will first start from $Y$ and describe a sequence of stochastic sampling operations to arrive at values for $v_j$. Think of these sampling operations as proceeding in the opposite direction as encoder inference, towards the input text.

We will need $i \in \{1 \ldots P\}$ which is an absolute position index that indexes across all possible positions in the size of the data window, $P$. We will also need $-P < r < P$, a relative position index.

1. $t_i \sim p(t_i)$, sample a token from a distribution.

2. $t_j \sim p(t_j|t_i)$ sample a second token from a distribution conditional on the choice of the first token.

3. $i \sim \text{uniform}(i)$, sample a position for $t_i$ from a uniform distribution over positions. The weights of this distribution are uniform, because as we shift the data across the window, the token $t_i$ will be observed equally often at every possible absolute position, $i$.

4. $r \sim p(r|t_i, t_j)$, sample a relative position for $t_j$ from a conditional distribution that depends on the two tokens involved.

5. $v \sim p(v|j)$: $p(v|j)$ is the prior on values in an embedding vector. For simplicity, we start with a categorical distribution. We discuss how this can be represented as a dense embedding vector in appendix E.

If we only want to talk in terms of absolute positions, we can introduce an absolute position index $j$, such that $j = i + r$.

$p(t_j|t_i)$ corresponds to $q_i \cdot k_j$ in the vanilla attention matrix. It has become commonplace to include a prior in this distribution $p(t_j|t_i, s)p(s)$ in order to design sparse attention patterns that achieve $O(n \log n)$ or $O(n\sqrt{n})$ complexity (Tay et al., 2020; Zaheer et al., 2020) (Costa et al., 2023). Additionally there are more recent grammatical attention patterns that achieve a sentence-level inductive bias towards grammatical constructions (Sartran et al., 2022). Although certainly useful, these priors on the attention weights are distinct from the additional inductive bias provided by the inductive transformer in the feed-forward layers of the transformer.

We have multiple redundant paths through this production that generate the same token at the same position. To transform this open-universe production into a closed-universe model, we will need to merge the probabilities of such multiple pathways, just as we have done whenever we have used an open-closed universe factor.

An easy check shows that this production does not match the vanilla transformer. In this production, for each possible pair $(t_i, t_j)$, there is a distribution over relative positions with a weight for every position. In other words, the number of learned weights is $O(E \cdot E \cdot P)$, where E is the embedding dimension and P is the size of the data window. By contrast, the number of learned weights in the vanilla attention mechanism is $O(E \cdot E \cdot 3)$. Why the discrepancy?

Let us look at entries in the vanilla attention matrix, $q_i \cdot k_j^T$, where $q_i$ and $k_j$ are attention vectors from positions $i$ and $j$. In the vanilla transformer, if we choose particular tokens $t_i$ and $t_j$, and only vary the position $j$, then the $(i, j)$-th entry in the attention matrix will decrease as position $j$ is more physically distant from position $i$ in the data window.

On the other hand, If we choose particular positions $i$ and $j$ as well as token $t_i$, and then only vary the token $t_j$, the weight will decrease as token $t_j$ is more "semantically distant" from $t_i$.

The vanilla transformer packs both the token embedding and the position embedding into the same embedding vector by adding a sinusoidal position vector to the semantic embedding vector. The blessing of dimensionality says that vectors in a high dimensional space are extremely likely to be linearly separable from one another (Gorban & Tyukin, 2018). This means that the semantic and spatial contributions to the embedding vector can operate almost independently of one another. By collapsing the position and semantic weights into a single representation, however, the vanilla transformer essentially performs approximate inference on the production, rather than exact marginalization, thereby reducing the complexity in the form of fewer learned weights.

Although our production requires $P$ times more weights, it more exactly articulates the deep motivation of the attention mechanism - shift invariance. By surfacing this underlying mechanism, it becomes possible to derive alternative implementations. There are likely to be new ways to introduce inductive bias into the shift invariant production that simplify the complexity of marginalization while also enhancing the identifiability and controllability of the model, and yielding improved scaling laws.

That said, there is nothing to prevent us from using a vanilla attention mechanism within an inductive transformer! And it would still be an inductive transformer, because of the benefits of the inductive bias introduced outside of the attention matrix. This can be an attractive option if the goal is to implement an inductive transformer with minimum architectural changes.

### C.3    A NOTE ABOUT CREDENCES AND PROBABILITIES

We have understood the attention mechanism in terms of steps in a generative production that samples symbolic (discrete) random variables. Marginalization (or other approximate inference) can be used to estimate the uncertainties or credences of these variables.

Why would we expect a large language model to be estimating uncertainty about discrete variables? The reason is that these systems are solving an inverse problem. If you want to guess what concepts

I am thinking as I say something to you, so that you can model my concepts and reply appropriately, then the operation I performed which was to transform my concepts into speech must be inverted by you who now must (approximately) convert my speech back into concepts. Since the mapping from concepts to speech is many-to-many, in listening to me, you have an inherently under-determined problem to solve, which by its nature requires representing the uncertainty of competing interpretations of the data. That said, presumably I am certain about what I am thinking as I talk about it. In other words there are some underlying, highly certain, symbolic (discrete) variables in my mind that you need to estimate. This is the same setup as in any digital communications or storage system. The transmitted messages are digital symbols and the receiver must perform marginalization in a model in order to arrive at the most likely guess for what bits were transmitted (Wymeersch, 2007).

Should we expect the system to represent uncertainty as proper probabilities? It is not at all un-common in machine learning systems for some operations to occur in the probability domain, some in the log probability domain, and even to perform operations in a domain that might best be de-scribed as a second order Taylor series approximation to the log probability domain. For example, above-threshold CMOS analog implementations of neural networks and probabilistic models take this approach, since the transfer function of these transistors is $I \propto V^2$ (Vigoda, 2003). Further-more, since taking a log is monotonic, and since during training we care mostly about gradients, the choice of domain generally does not materially impact the expressiveness of the model. In the vanilla transformer architecture, there are places like the output softmax that treat uncertainties as log probabilities to be exponentiated. There are other places, like the attention matrix, that seem to treat activations as probabilities that can be multiplied. The enormous generality of the multilayer perceptrons allows for transformations between representations, as well as varying representations at different locations in the network.

## D $\wedge$ IN THE LOG PROBABILITY DOMAIN

After the feed-forward layer, the vanilla transformer has an add & norm layer that adds in the resid-ual connections and normalizes the output of the layer. Our layer of $\wedge$ activation functions in figure 1 performs the analogous operation in the inductive encoder. Recall, from equation 46 that the prob-ability $p(z = 1)$ is given by the product of the probabilities $p(x)$ and $p(y)$. In the log domain this results in,

$$\log p(z = 1) = \log p(x = 1) + \log p(y = 1). \tag{50}$$

As we normalize after computing this activation function, this exactly mirrors the behavior of the "Add & Normalize" layer in the vanilla transformer.

Note that the vanilla encoder has an additional add & norm layer after the feed-forward layer. We would argue that the sum with residuals here is overkill, but that the layer norm does correspond to the Bernoulli to categorical that happens after the encoder close-open-universe factor.

## E FROM INDICATOR VECTORS TO EMBEDDING VECTORS

Until now we have been assuming embedding vectors that represent the probabilities of states in a categorical distribution. Categorical vectors are not an efficient embedding, but they are easy to understand. They are of length $T$, the number of tokens in the vocabulary, indexed by $t \in \{1, \ldots, T\}$. Each element in these categorical vectors represents $p(t)$, the probability of the $t^{\text{th}}$ token. Just as in the vanilla transformer, categorical embeddings can be used to represent not just tokens but all kinds of information deep within a network. Throughout this paper we have done just that. The inner product of two indicator vectors is meaningful as a measure of the distance between two categorical distributions, so we can compute semantic distance just as we would with dense embedding vectors.

As in the vanilla transformer, at the input of the inductive transformer we have an embedding vector for each token position in order to represent the token that is present at that position. If, as is true at the input to the network, we are certain about what token is present at a given position, then we have a "one-hot" vector with a single probability that is $\approx 1$ for the $t^{\text{th}}$ token and $\approx 0$ elsewhere. If, as is true at the output of the network, we have a categorical distribution over tokens, then we have

an array of probabilities from which we can sample a token. The inputs and outputs are therefore identical to those of the vanilla transformer.

In order to implement *dense* embedding representations within the inductive transformer, we do the typical thing and use an embedding matrix $E$ to transform a categorical vector to a lower dimensional "dense" embedding vector,

$$p(i_{\text{embedding}}) = E \cdot p(i_{\text{categorical}}) \tag{51}$$

$$= \sum_{i_{\text{categorical}}} p(i_{\text{embedding}}|i_{\text{categorical}})p(i_{\text{categorical}}), \tag{52}$$

and then at the output use $E^T$ to transform from an embedding vector back to a categorical vector. The only problem with this approach for the inductive transformer is that all of our factors were defined to operate on categorical variables and/or to transform a categorical variable to a group of Bernoulli variables, operate on Bernoullis, and then transform back to categorical.

Constructing an inductive transformer with dense embedding vectors is not present a major challenge however. Assume for example, an arbitrary factor where the variables $a_{\text{categorical}}, b_{\text{categorical}}, c_{\text{categorical}}$ are categorical. As always, inference is marginalization. For example,

$$p(a) = \sum_{b,c} p(a|b,c)p(b)p(c) \tag{53}$$

If we only know $p(a|b,c)$ in the categorical representation, we can always use our learned $E^T$ to project our incoming distributions from the dense embedding vector representation into categorical representation,

$$p(b_{\text{categorical}}) = \sum_{b_{\text{embedding}}} p(b_{\text{categorical}}|b_{\text{embedding}})p(b_{\text{embedding}}) \tag{54}$$

$$p(c_{\text{categorical}}) = \sum_{c_{\text{embedding}}} p(c_{\text{categorical}}|c_{\text{embedding}})p(c_{\text{embedding}}), \tag{55}$$

then apply the factor in categorical representation as we did in equation 53, and then project back again using $E$,

$$p(a_{\text{embedding}}) = \sum_{a_e} p(a_{\text{embedding}}|a_{\text{categorical}})p(a_{\text{categorical}}). \tag{56}$$

The only problem with this is that the categorical representation may be very large, making this computationally inefficient. We can solve this problem by training a multi-layer perceptron to approximate the entire function from dense embeddings to dense embedding, while remaining in a smaller number of dimensions the entire time. This is, in fact, what we would argue is happening in the vanilla transformer.

Using categorical variables in our exposition, however, allowed us the theoretical and pedagogical luxury of more easily explaining the factors in the inductive transformer.

