# OpenReview forum: "Inductive Transformers: How Large Language Models Form Concepts, And How to Make Them Even Better At It"
_ICLR.cc/2024/Conference — Submitted to ICLR 2024_

### Official Review · Reviewer_onZM · 2023-10-31

**Soundness:** 2 fair
**Presentation:** 1 poor
**Contribution:** 2 fair
**Rating:** 3
**Confidence:** 3

**Summary:**

This work provides a re-interpretation of transformers as a probabilistic generative model of concepts and presents a new architecture, inductive transformers, that enables a stronger inductive bias towards more structured concepts. Tokens are encoded into latent variables with their own hierarchies/relations with other latent variables, and a decoder maps these latent variables back into tokens. Much machinery of the model is the same as vanilla transformers, but the operations are cleverly recast into this probabilistic framework, but there are also many additions are made to the model to accommodate this probabilistic framework. They trained their model on a toy dataset with two-token pairs and show that it can learn abstract concepts like modifiers ("big cat" or "big dog"), puts abstract concepts consistently in the same location within the model, and exhibits controlability in being able to delete concepts. This shows a potential improvement to transformers in terms of interpretability.

**Strengths:**

* The re-casting of transformers into a probabilistic framework is interesting and clever.

* The toy experiments shows the model has interesting properties (controllability, etc.) that can improve transformers more.

* They tackle a well-motivated problem in how to build structured concepts into transformer networks.

**Weaknesses:**

* The paper is severely lacking clarity. Most of the details needed to understand the model are hidden in the appendix. Its not at all clear how this model works without digging through the appendix, and even then it takes a long time to figure out how the different components come together. This is especially puzzling given that the authors have so much extra space left in the main paper. The introduction, I feel, can be condensed a lot and the content in the supplement should be moved into the main paper such that readers of the main paper should understand how this architecture is implemented.

* The model is only implemented on a very simple toy dataset (two word sentences). It would be better to see how this model can be used on a larger dataset that is more on the level of what transformers are trained on.

**Questions:**

* A lot of components of the model seem highly similar to clone structured cognitive graphs (George et al. 2021). What are the similarities/differences to their work?

References:

George, D., Rikhye, R. V., Gothoskar, N., Guntupalli, J. S., Dedieu, A., & Lázaro-Gredilla, M. (2021). Clone-structured graph representations enable flexible learning and vicarious evaluation of cognitive maps. Nature communications, 12(1), 2392.

---

> ### Author Response · Authors · 2023-11-10
> **Thank you; Request for link to Patel et. al. on CNNs**
>
> Thank you so much for all the thoughtful reviews! We are very grateful for your time, and have begun addressing each of the suggestions!
>
> We would like to incorporate all of your suggested prior art.  One reviewer suggested Patel et. al. on CNNs.  Would you be able to provide a link or a little more specific information to find that source?
>
> Thank you so much and all the best!

---

> > ### Comment · Reviewer_onZM · 2023-11-14
> >
> > I believe reviewer Yhkg is the one that brought up Patel et al., not me (I am onZM).
> >
> > Also please do not mention your first names. We are supposed to not know the authors' identities.

---

> ### Author Response · Authors · 2023-11-23
>
> Thank you so much, for your thoughtful comments. We address each of them below. Our comments have "-->"'s:
>
> Strengths:
> The re-casting of transformers into a probabilistic framework is interesting and clever.
> --> Thank you!
>
> The toy experiments shows the model has interesting properties (controllability, etc.) that can improve transformers more.
> --> Thank you!
>
> They tackle a well-motivated problem in how to build structured concepts into transformer networks.
> --> We agree!
>
> Weaknesses:
> The paper is severely lacking clarity. Most of the details needed to understand the model are hidden in the appendix. Its not at all clear how this model works without digging through the appendix, and even then it takes a long time to figure out how the different components come together.
>
> --> We have now completed a lot of edits to hopefully make the paper more accessible. We agree that it would have been nice to be able to include (a) a high-level figure for the production and statistical model without all of the factor graph details, (b) a comparison table with more space to explain the comparison of each layer type (c ) the factored probability distribution for vanilla and for inductive transformers. (The factored probability distributions for the vanilla and for inductive transformers are actually identical to one another, so it would be the same equation twice.) (4) inline much of the appendices.  In the archival version, where we are afforded more space, we will make these changes.
>
> This is especially puzzling given that the authors have so much extra space left in the main paper.
> --> We checked into this and confirmed that in ICLR we are limited to 9 pages for the main body of the paper.  We are right up against the limit.
>
> The introduction, I feel, can be condensed a lot
> -->  We actually assembled a draft exactly like this, and to be honest it really did not seem to help.  Bringing in slivers of appendices into the main body seemed to be both too little additional explanation and also too much detail, both at the same time.  Even if we eliminated the introduction almost entirely, it would not afford enough space for all but a sliver of the appendices.
>
> and the content in the supplement should be moved into the main paper such that readers of the main paper should understand how this architecture is implemented.
>
> --> We completely agree! The content in the supplement should be moved into the main paper. In an archival version where space allows, we will do that.  Unfortunately, ICLR severely limits the available space to a strict limit of 9 pages.  That said, we do feel that the reader who hopefully gains from the main body a high-level appreciation of the story, can open the appendices side by side with the main body and refer back and forth fairly efficiently.  In other words, all of the supporting material is, at the end of the day, contained in the submission.
>
> --> Since the initial submission, we have also done a tremendous amount of further editing on both the main body and the appendices  in order to try to make them more understandable and to tie it all together more seamlessly.
>
>
> The model is only implemented on a very simple toy dataset (two word sentences). It would be better to see how this model can be used on a larger dataset that is more on the level of what transformers are trained on.
>
> --> Two thoughts here:
>
> (1) We agree that a key next step will be to scale this up! That said, we see it as future work. We intend this first paper to be analogous to theoretical physicists proposing a new mathematical model with simulations that show that the math works. Future work then typically involves a larger team of experimentalists. We have now joined with an experimental team to do just that, but we would humbly submit that this paper represents a publishable contribution in terms of new ideas, novel mathematical derivations, and code that implements the model and runs.
>
> (2) Scaling laws for large language models are critical for the field. That said, models scale up, but they do not scale down. There is a minimum data set size below which all previous transformer architectures do not learn anything. Ours does. We now state explicitly in the text that "We design a transformer that successfully learns even as the data set size scales to zero."
>
>
> Questions:
> A lot of components of the model seem highly similar to clone structured cognitive graphs (George et al. 2021). What are the similarities/differences to their work?
>
> --> My co-author read the work you suggested, and will provide an answer in an additional comment.
>
>
> Thank you again for your feedback!

---

> > ### Author Response · Authors · 2023-11-23
> >
> > Dear Reviewer,
> >
> > Regarding your question "A lot of components of the model seem highly similar to clone structured cognitive graphs (George et al. 2021). What are the similarities/differences to their work?"
> >
> > Thank you for pointing out this interesting read. There are indeed several similarities between this paper and ours.
> > In particular, clone-structured cognitive graphs (CSCGs) are focused on learning higher-order relational representations from sequence data. Similarly, we aim to organize higher-level abstract concepts in inductive transformers.
> > There are also several notable differences:
> > 1. The training methodology differs significantly - we introduce custom activation functions, connectivity patterns, and synthetic training data.
> > 2. Inductive transformers have direct ties to the underlying transformer architecture. CSCGs seem more standalone, or in this paper more closely tied to biology.
> > 3. Inductive transformers connect model form to recursive marginalization in transformers. CSCGs have a domain-specific probabilistic graph formulation.
> > 4. This leads to very different mathematical groundings for the two approaches.
> >
> > Let us know if there is any point you'd like us to further clarify.
> >
> > Thank you again for your feedback!

---

### Official Review · Reviewer_x6Rf · 2023-11-01

**Soundness:** 2 fair
**Presentation:** 1 poor
**Contribution:** 2 fair
**Rating:** 3
**Confidence:** 2

**Summary:**

The paper present a new approach to inject additional inductive bias into transformers to enable tighter conceptual organization, greater conceptual control, and higher levels of conceptual abstraction. The approach is given an illustrative example simulation.

**Strengths:**

The paper propose a generative statistical model such that recursive marginalization of athe model is in tight equivalence with the calculations performed by inference in a vanilla transformer. This idea can provide a foundation for the design of new inductive bias into transformers, which is inductive transformers.

**Weaknesses:**

It's very hard to understand the equivalence between the proposed generative model and the vanilla transformer.

**Questions:**

1. It's difficult to understand the equivalence between the generative model and vanilla transformer, is there more formal way to prove the equivalence?
2. Can you provide more explanations on the "concept" meaning in both the generative model and vanilla transformer.
3. What's the time complexity of the propose generative model's training/inference process?
4. Is it possible to apply the proposed generative model on large scale dataset?

---

> ### Author Response · Authors · 2023-11-23
>
> Thank you so much, for your very thoughtful comments and questions.  We would like to speak to each of them here. Our comments have "-->"'s:
>
> Strengths:
> The paper propose a generative statistical model such that recursive marginalization of athe model is in tight equivalence with the calculations performed by inference in a vanilla transformer. This idea can provide a foundation for the design of new inductive bias into transformers, which is inductive transformers.
>
> --> yes exactly! we are on the same page here.
>
>
> Weaknesses:
> It's very hard to understand the equivalence between the proposed generative model and the vanilla transformer.
>
> --> We completed a lot of edits to hopefully make the paper more accessible   We agree that it would have been nice to be able to include (a) a high-level figure for the production and statistical model without all of the factor graph details, (b) a comparison table with more space to explain the comparison of each layer type (c ) the factored probability distribution for vanilla and for inductive transformers.  (The factored probability distributions for the vanilla and for inductive transformers are actually identical to one another, so it would be the same equation twice.)  In the archival version, where we are afforded more space, we can include these.
>
> Questions:
> It's difficult to understand the equivalence between the generative model and vanilla transformer, is there more formal way to prove the equivalence?
>
> --> One potential way we thought of to show this in more formal detail is with the code.  In our code, we show the pytorch calculations for our layers and we bring in the Harvard code for the "annotated transformer" which is an implementation of the vanilla transformer.  This makes the the comparison very apples-to-apples and the equivalence becomes readily apparent.
>
> --> Another way is to see that the factorization of the probability distribution is identical.
>
> --> Between the higher level representation of the probability notation and the lowest level which is the code, we have not identified an intermediate representation that would allow us to make the equivalence more formally and pedagogically explicit.   Wer are very open to suggestions!
>
> Can you provide more explanations on the "concept" meaning in both the generative model and vanilla transformer.
>
> --> In response to your comment, we added this explanation: "We define a ``concept'' as a sub-network that can generate many different but synonymous token sequences (e.g. ``tiny canine'')."
>
> What's the time complexity of the propose generative model's training/inference process?
>
> --> In response to your comment, we added this explanation: "The inductive transfomer is a more focused version of the vanilla transformer, and will therefore generalize similarly.  The time and space complexity is identical."
>
>
> Is it possible to apply the proposed generative model on a large-scale dataset?
>
> -->  Two thoughts here:
>
> (1) We agree that a key next step will be to scale this up!  That said, we see it as future work. We intend this first paper to be analogous to theoretical physicists proposing a new mathematical model with simulations that show that the math works. Future work then typically involves a larger team of experimentalists. We have now joined with an experimental team to do just that, but we would humbly submit that this paper represents a publishable contribution in terms of new ideas, novel mathematical derivations, and code that implements the model and runs.
>
> (2) Scaling laws for large language models are critical for the field. That said, models scale up, but they do not scale down. There is a minimum data set size below which all previous transformer architectures do not learn anything. Ours does. We now state explicitly in the text that "We design a transformer that successfully learns even as the data set size scales to zero."
>
> Thank you again for your feedback!

---

### Official Review · Reviewer_Yhkg · 2023-11-01

**Soundness:** 2 fair
**Presentation:** 1 poor
**Contribution:** 1 poor
**Rating:** 1
**Confidence:** 5

**Summary:**

I had a hard time understanding the core contribution of the paper. As far as I can tell, the architecture is a sort of "parallel grammar" where abstract productions are learned and used to generate concrete tokens, in the same way that a PCFG might generate terminal symbols along with a complete abstract structure supporting it.  The vanilla transformer is viewed as the marginalization of this model. The hope is that by incorporating / learning abstract structure along with concrete productions, the model will have a more powerful inductive bias.

**Strengths:**

The high-level goal of the paper, which is to create transformers with strong, hierarchical and compositional inductive biases, is a great research direction.

The idea that vanilla transformers can be interpreted as the marginalization of a structured probabilistic model is intriguing; similar work has been done by Patel et al on CNNs.

**Weaknesses:**

While the high-level goals are laudable, the paper suffers from several weaknesses:

- The paper does a poor job of explaining its ideas. While the introduction is reasonably accessible, I struggled to understand the probabilistic model, the production system, and the need for Bernoulli/categorical variables. I feel like this could have been framed much more clearly using terminology / frameworks that are more common in the machine learning community.  For example, this could have been framed in terms of probabilistic graphical models, or graph grammars, etc.

- The empirical results were unconvincing. The training set was too simplistic, and the simple lesion done does not confirm the hypothesis of structured / compositional knowledge.

**Questions:**

none

---

> ### Author Response · Authors · 2023-11-23
>
> Thank you so much for your comments! We would like to go through each of them here. Our comments have "-->"'s:
>
> Strengths:
> The high-level goal of the paper, which is to create transformers with strong, hierarchical and compositional inductive biases, is a great research direction.
> --> Thank you, we agree!  We would love to discuss this more with you at the conference if we get in!
>
> The idea that vanilla transformers can be interpreted as the marginalization of a structured probabilistic model is intriguing; similar work has been done by Patel et al on CNNs.
> --> Patel is a common name, so we were not able to find this particular paper.  Feel free to provide it and we will include a citation.  Your comment shows that you clearly appreciate what we are trying to do!
>
> Weaknesses:
> While the high-level goals are laudable, the paper suffers from several weaknesses:
>
> The paper does a poor job of explaining its ideas. While the introduction is reasonably accessible, I struggled to understand the probabilistic model, the production system, and the need for Bernoulli/categorical variables.
>
> --> We have now done a tremendous amount of edits and revisions to hopefully make the paper more accessible.
>
>
> I feel like this could have been framed much more clearly using terminology / frameworks that are more common in the machine learning community. For example, this could have been framed in terms of probabilistic graphical models, or graph grammars, etc.
>
> --> We actually explicitly tried to avoid this.  Although it might make the paper more accessible to a specific sub-field with the particular vocabulary, many in deep learning are not familiar with probabilistic graphical models, belief propagation, graph grammars, etc.  We have been trying to restrict ourselves to only using simple probability theory and self-contained explanations.
>
> The empirical results were unconvincing. The training set was too simplistic, and the simple lesion done does not confirm the hypothesis of structured / compositional knowledge.
>
> --> Three thoughts here:
>
> (1) We also agree that a key step will be to scale this up! We see that as future work. We intend this first paper to be analogous to theoretical physicists proposing a new mathematical model with simulations that show that the math works. Future work then typically involves a larger team of experimentalists. We have now joined with an experimental team to do just that, but we would humbly submit that this paper represents a publishable contribution in terms of new ideas, novel mathematical derivations, and code that implements the model and runs.
>
> (2) Scaling laws for large language models are critical for the field. That said, models scale up, but they do not scale down. There is a minimum data set size below which all previous transformer architectures do not learn anything. Ours does. We now state explicitly in the text that "We design a transformer that successfully learns even as the data set size scales to zero."
>
> (3) This paper essentially shows how to efficiently learn a PCFG using belief propagation as inference and back-prop for learning.  For the first time PCFG structure learning does not require using inside-outside, E-M, and all that.  We do it with tensors in PyTorch.  The literature on grammar learning at smaller scales and less scalable algorithms, however, does demonstrate that learned PCFGs are structured and compositional, and we cite, for example, Goodman et. al. who have demonstrated machine learning of compositional and structural PCFGs to grade student programming assignments.  Space did not allow for a full treatment of grammar learning literature.   For the deep learning reader, we show the startling conclusion that the vanilla transformer actually IS a grammar learner.  Seen from that perspective, this work in some sense unifies PCFG learning and transformers into a single mathematical framework.

---

### Official Review · Reviewer_XTyg · 2023-11-05

**Soundness:** 2 fair
**Presentation:** 3 good
**Contribution:** 3 good
**Rating:** 5
**Confidence:** 2

**Summary:**

This paper proposes a way of incorporating inductive biases into the design of a vanilla transformer in a minimally invasive way: in the form of the activation functions and connections in the model. This is motivated by drawing comparisons with how humans are able to learn controllable abstract organised concepts and perform causal reasoning as a result. Successful variants of such a model might pave the way forward for models that are smaller and more capable than existing general models trained with large amounts of data and using massive computational resources.

**Strengths:**

- In my opinion, the key strength of the paper is the contribution of a straightforward model whose motivation is to achieve improved causal reasoning, iterative experimentation, long-range planning, and other cognitive abilities in language models, which are active research areas.
- The illustrative example is intuitive and helpful to get a sense of all the advantages of the inductive transformer elicited in the paper.
- The definitions for key ideas like identifiability, controllability are clear, increasing the readability of the paper.

**Weaknesses:**

- While the paper provides an illustrative example to demonstrate the inductive transformer's potential, it lacks empirical evaluation with large-scale experiments and/or diverse datasets, which reduces its impact in my opinion. A more comprehensive evaluation would provide a clearer understanding of its capabilities and limitations.
- The introduction of additional inductive biases could potentially result in an increased model complexity and training cost. It would be helpful to discuss the trade-offs between model performance and computational resources, as well as practical scalability issues that may arise.
- The paper focuses on concept learning, but does not discuss in detail the interpretability and explainability of the inductive transformer. Understanding how this model forms concepts and making its decision-making processes more interpretable is important in assessing its utility.

**Questions:**

- If we agree with the premise that inductive transformers enable learning of abstract concepts, how would we go about generalising these concepts to a wide range of tasks and domains? Each task/domain might warrant different inductive biases being incorporated into the model architecture, resulting in a complete loss of generality from one task to another. Would that be the case?
- Could you discuss a potential merge of base models that have been trained to perform next token prediction and the minimal architectural changes to the transformer that result in the inductive transformer? Or would the proposed architecture entail completely getting rid of base models?

---

> ### Author Response · Authors · 2023-11-23
>
> We deeply appreciate your thoughtful comments!  We would like to speak to them one by one.  Our comments have "-->"'s:
>
> Strengths:
>
> ... The key strength of the paper is the contribution of ... a model whose motivation is to achieve improved causal reasoning, iterative experimentation, long-range planning, and other cognitive abilities in language models, which are active research areas.
>
> --> We agree!  Many would also agree that without additional inductive bias, and with only the current scaling path for large language models with larger multimodal datasets and RLHF, it could be extremely expensive to achieve these capabilities, and may not be feasible.  We actually changed the discussion section to list the contributions of the paper:
>
> "This paper offers the following contributions: (1) We provide the first demonstration of causal intervention in a transformer model. For example, we show how to delete specific concepts by deleting specific sub-networks. (2) We design a transformer that successfully learns even as the data set size scales to zero. (3) We design a transformer such that the concepts it learns are localized within
> identifiable sub-networks. (4) We show that the feed-forward layers of a vanilla transformer learn underlying functions that can instead be derived analytically. (5) We derive from first principles why the multi-layer perceptrons in the feed-forward layer of the vanilla transformer are factored the way they are. (6) We show that the connectivity from the encoder to the decoder in the vanilla transformer is not correct and how to fix it.2 (8) We derive the testable prediction that training data with a particular inductive bias can help unlock a range of important new abilities for large language models, including curiosity. One could generate synthetic data from, for example, the model we described here, and use this synthetic data within the overall mix of training data for a foundation model (Aky ̈urek et al., 2020). (9) We show that this inductive bias training data can be replaced or augmented by directly designing the inductive bias into the activation functions and connectivity of the model. (9) We mathematically define concepts, and explain why scaling up models yields greater
> conceptual abstraction. We suggest that deeper abstractions manifest as “emergent” capabilities."
>
> The illustrative example is intuitive and helpful to get a sense of all the advantages of the inductive transformer elicited in the paper.
> The definitions for key ideas like identifiability, controllability are clear, increasing the readability of the paper.
> --> Thank you!
>
>
> 1. The paper lacks empirical evaluation with large-scale experiments and/or diverse datasets, which reduces its impact in my opinion. A more comprehensive evaluation would provide a clearer understanding of its capabilities and limitations.
>
> --> Two thoughts here:
>
> Scaling laws for large language models are critical for the field.  That said, models scale up, but they do not scale down.  There is a minimum data set size below which all previous transformer architectures do not learn anything.  Ours does.  We now state explicitly in the text that "We design a transformer that successfully learns even as the data set size scales to zero."
>
> We also agree that a key step will be to scale this up!  We see that as future work.   We intend this first paper to be analogous to theoretical physicists proposing a new mathematical model with simulations that show that the math works.  Future work then typically involves a larger team of experimentalists.  We have now joined with an experimental team to do just that, but we would humbly submit that this paper represents a publishable contribution in terms of new ideas, novel mathematical derivations, and code that implements the model and runs.
>
> The introduction of additional inductive biases could potentially result in an increased model complexity and training cost. It would be helpful to discuss the trade-offs between model performance and computational resources, as well as practical scalability issues that may arise.
> --> Inductive bias always reduces the size of the overall search space.  It can, however, make the search space more difficult to traverse which can be costly in compute.  That is not the case here.  As we show by reproducing the training run 100 times, the convergence is efficient and repeatable.  it does not, for example, get stuck in local minima.
>
> The paper focuses on concept learning, but does not discuss in detail the interpretability and explainability of the inductive transformer. Understanding how this model forms concepts and making its decision-making processes more interpretable is important in assessing its utility.
> --> The majority of ML readers are not focused on interpretability, but we feel that for those who are, the implications are clear from considering the controllability and identifiability sections.  Concepts in this model are localized and can be deleted.

---

### Meta-Review · Area_Chair_wf7w · 2023-12-06

**Metareview:**

The authors present ways to alter transformer inductive biases. Reviewers agree that the direction is valuable and super interesting, but found the paper lacking in clarity (understanding the PGM and its equivalence to transformers, what is a "concept" ) making it difficult to evaluate the contributions and take home message, and that the domain evaluated were too simplistic.

**Justification For Why Not Higher Score:**

but found the paper lacking in clarity (understanding the PGM and its equivalence to transformers, what is a "concept" ) making it difficult to evaluate the contributions and take home message, and that the domain evaluated were too simplistic.

**Justification For Why Not Lower Score:**

NA

---

### Decision · Program_Chairs · 2024-01-16

Reject